# WHEN MODALITIES CONFLICT: HOW UNIMODAL REASONING UNCERTAINTY GOVERNS PREFERENCE DYNAMICS IN MLLMS

## ABSTRACT

Multimodal large language models (MLLMs) must resolve conflicts when different modalities provide contradictory information, a process we term modality following. Prior work measured this behavior only with coarse dataset-level statistics, overlooking the influence of models' confidence in unimodal reasoning. In this paper, we introduce a new framework that decomposes modality following into two fundamental factors: relative reasoning uncertainty (the case-specific confidence gap between unimodal predictions) and inherent modality preference (a model's stable bias when uncertainties are balanced). To validate this framework, we evaluate a diverse suite of MLLMs on a novel controllable dataset and multiple real-world benchmarks. Using entropy as a fine-grained uncertainty metric, we uncover a universal law: the probability of following a modality decreases monotonically as its relative uncertainty increases. This allows us to identify the balance point, which represents the relative difficulty level where the model follows both modalities with comparable probability. Unlike traditional macro-level ratios, this measure offers a more principled and less confounded way to characterize modality bias, disentangling it from unimodal capabilities and dataset artifacts. Further, by probing layer-wise predictions, we reveal the internal mechanism of oscillation: in ambiguous regions near the balance point, models vacillate between modalities specifically in the middle-to-late layers, explaining externally observed indecision. Finally, we demonstrate the practical utility of our framework for preference steering. We show that data efficiency is governed by relative uncertainty: training on "easy" samples fails to generalize, whereas targeting boundary cases identified by our metric is essential for robust preference alignment.

## 1 INTRODUCTION

Multimodal large language models (MLLMs) (Achiam et al., 2023; Team et al., 2023; Wang et al., 2024; Yin et al., 2024; OpenAI et al., 2024) demonstrate powerful capabilities by processing information from various sources, like images and text, making them vital in applications ranging from web navigation (OpenAI, 2025) to aiding visually impaired users. However, a critical challenge arises when these modalities present conflicting information. For example, an image might show a blue car, while the accompanying text describes it as red. In such cases, the MLLM must resolve the conflict, leading to an observable behavior we term **modality following**: the model's final output aligns with the information from one modality over the other.

Prior studies (Zhang et al., 2025; Deng et al., 2025) have typically examined this phenomenon using coarse, dataset-level statistic: the ratio of text-following versus vision-following cases on a given set of conflicting inputs. This approach, however, often attempts to neutralize the model's unimodal capabilities by filtering for cases where the model can correctly answer based on either modality alone. This overlooks a crucial factor: the model's *confidence* in each of its unimodal predictions. For the same instance, one model may produce the correct answer with high confidence while another does so with low confidence. Even within a single model, two different instances can elicit correct unimodal answers but with vastly different certainty levels. Such variations in underlying confidence directly influence the model's final choice in multimodal settings and, consequently, shape the aggregate statistics of modality-following behavior.

To truly understand the modality-following process, we propose that the static, dataset-level following statistics are emergent properties of two distinct underlying factors: (1) the **relative reasoning uncertainty** between the two modalities on a case-by-case basis, measured under unimodal inputs, which reflects the model's confidence gap between text-only and vision-only reasoning, and (2) a more stable, **inherent modality preference**, which we define as the model's intrinsic leaning toward one modality when the reasoning uncertainties from both are perceived as equal. This leads to our central hypothesis: **An MLLM's modality-following behavior is a dynamic process governed by the interplay between the relative reasoning uncertainty of the conflicting modalities and the model's own inherent preference.** In simpler terms, a model's decision to follow the text depends on whether the text's reasoning advantage (i.e., its low relative uncertainty compared to the image) is significant enough to overcome the model's potential inherent preference for vision.

We quantified the model's perceived uncertainty for each unimodal case using the *output entropy* of its answer token, where a higher value indicates lower confidence (Shannon, 1948; Farquhar et al., 2024; Zhang et al., 2024a; Cao & Ou, 2025).Our overall analysis process is shown in Figure 1. To validate the hypothesis, we constructed a controllable toy dataset that allows us to systematically and independently manipulate the reasoning difficulty of visual and textual inputs, thereby inducing varying levels of uncertainty in unimodal reasoning. The relationship between these two uncertainty scores was then used to define the relative uncertainty, forming the central axis for our analysis.

Our first goal was to verify if relative uncertainty indeed governs the model's final choice. By analyzing the model's outputs across our benchmark, we uncovered a clear and predictable pattern. As we systematically increased the reasoning uncertainty of one modality relative to the other, the model's probability of following that modality showed a consistent **monotonic decrease**. This finding confirms that modality following is not a fixed attribute but a fluid behavior that predictably shifts with the relative difficulty of unimodal inputs. However, we observed that a model does not necessarily follow the modality with the lower relative uncertainty. Instead, each model possesses a unique threshold—a subjective **balance point** of uncertainty that it is willing to tolerate. This balance point reveals the model's **inherent preference**. For example, a model with a strong inherent preference for vision might only follow the text if the text is *significantly* easier to process than the image.

Having established this behavioral relationship, we then sought to understand the internal mechanism behind it. *Why* does a model hesitate or average its choices when the relative uncertainty is near its subjective balance point? To explore this, we categorized conflict scenarios into two types: a **clear region**, where one modality is significantly dominant, and an **ambiguous region**, where uncertainties are balanced. By probing layer-wise predictions, we reveal that the model's hesitation is visible internally as **"oscillations"**, where the top prediction repeatedly switches between conflicting modalities. Crucially, we discover that this cognitive struggle is not uniform but strictly localized to the middle-to-late layers. While early layers process features relatively stably, the model repeatedly vacillates in these deeper layers when facing ambiguity, directly explaining the externally observed indecision.

Finally, we apply our framework to the practical challenge of **preference steering** via Supervised Fine-Tuning (SFT). Our experiments reveal a critical "failure of easy-to-hard generalization": models trained solely on data where the target modality is already dominant ("easy" data) fail to learn robust modality following. We demonstrate that effective steering requires targeting the **boundary cases** identified by our relative uncertainty metric, providing a principled guideline for data selection in alignment tasks. In summary, this paper makes four key contributions:

- We decompose modality following behavior into two core components: case-specific **reasoning uncertainty** and the model's stable **inherent preference**.
- Leveraging both **novel controllable** and **real-world datasets**, we establish a fundamental law: modality following probability monotonically decreases as reasoning uncertainty increases, allowing us to quantify inherent preference as the **balance point**.
- We uncover an internal "oscillation" mechanism localized to **deeper layers**, revealing that conflict resolution is a high-level process linking internal dynamics to external hesitation.
- We apply our framework to preference steering via finetuning, demonstrating that data efficiency is governed by reasoning uncertainty, thereby offering concrete guidance for future data selection.

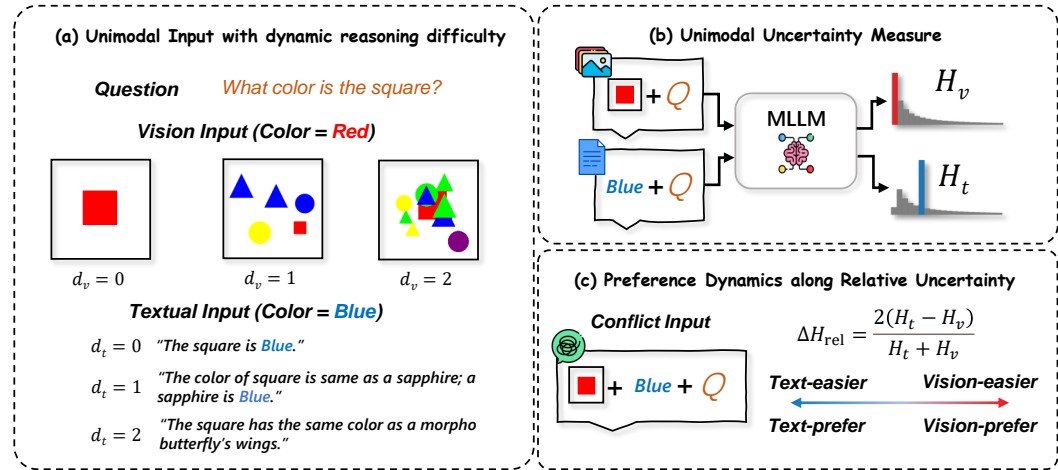

Figure 1: Overview of the analytical framework. **(a)** We create inputs with independently controllable visual ($d_v$) and textual ($d_t$) difficulty. **(b)** We measure the model's perceived uncertainty for each modality via output entropy ($H_v$, $H_t$). **(c)** We then use the relative uncertainty ($\Delta H_{rel}$) to analyze the model's choice when faced with a conflict.

## 2 DEFINING CONFLICTING INPUTS AND QUANTIFYING MODALITY FOLLOWING

**Conflicting Inputs.** We define a *conflicting input* as a triplet $(I, T, Q)$ consisting of an image $I$, a textual description $T$, and a question $Q$, such that the unimodal predictions of the MLLM $M_\theta$ disagree:

$$Y_v = M_\theta(Q, I) \neq Y_t = M_\theta(Q, T).$$

Here, $Y_v$ and $Y_t$ denote the predictions when the model relies solely on the visual or textual modality, respectively. For example in Figure 1 (a), consider the question $Q$ = "What is the color of the square?". If the image $I$ shows a red square, while the text $T$ states "The color of the square is the same as a morpho butterfly's wings", then the image supports the answer "red" whereas the text suggests "blue". This forms a concrete instance of a conflicting input triplet $(I, T, Q)$. This setting requires the model to resolve contradictory cues and implicitly decide which modality to follow.

**Macro-level Metrics for Modality Following.** Given a conflicting input $x = (I, T, Q)$, the multimodal prediction is $Y_m = M_\theta(x)$. We categorize the outcome as **vision-following** if $Y_m = Y_v$, **text-following** if $Y_m = Y_t$, and **other** otherwise. To quantify the aggregate modality-following behavior on a dataset, we adopt the traditional approach of calculating following ratios. We define the text-following ratio (TFR) and vision-following ratio (VFR) as:

$$\text{TFR} = \frac{|\{x : Y_m = Y_t\}|}{|\{x : Y_m \in \{Y_v, Y_t\}\}|}, \quad \text{VFR} = 1 - \text{TFR}.$$

These ratios offer a simple, macro-level statistic of a model's aggregate behavior. In subsequent sections, we will deconstruct how these statistics emerge from a deeper interplay between case-specific uncertainty and a model's inherent preference, which these ratios alone cannot capture.

## 3 PREPARING FOR THE ANALYSIS: A CONTROLLABLE DATASET AND AN UNCERTAINTY METRIC

To systematically investigate our central hypothesis: **that modality following is governed by relative uncertainty and inherent preference**, we must first establish a controlled experimental setup. This section details the two essential preparations for our analysis: (1) the construction of a novel dataset with independently controllable difficulty levels for both vision and text, and (2) the validation of entropy as the uncertainty metric, to precisely quantify the model's perceived reasoning difficulty in a fine-grained, modality-comparable manner.

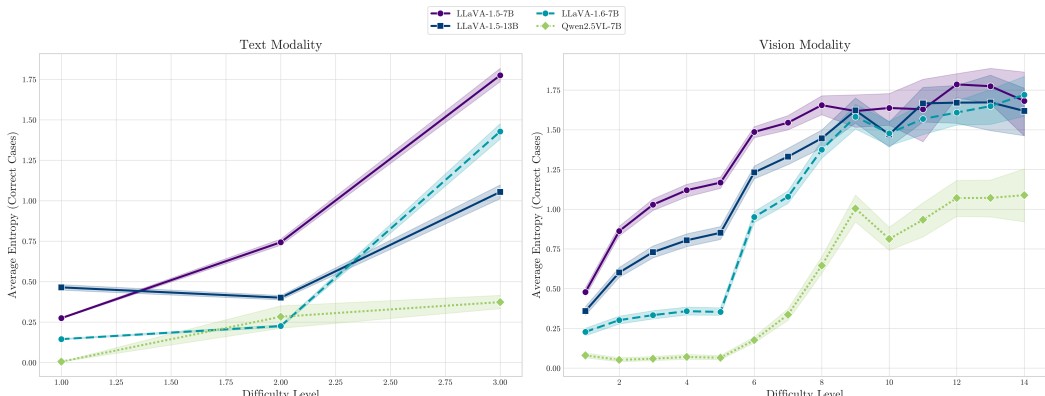

Figure 2: Unimodal Entropy Trends Across Difficulty Tiers. Average unimodal entropy for text (left) and vision (right) as a function of our designed difficulty tiers. Across all models, entropy consistently increases with difficulty, validating its use as a proxy for model-perceived uncertainty and revealing differences in model capabilities.

### 3.1 CONSTRUCTING A DATASET WITH CONTROLLABLE DIFFICULTY

Existing benchmarks lack the ability to systematically vary the reasoning difficulty of each modality independently. To overcome this, we built a toy dataset where each multimodal instance is defined by a task type $\mathcal{T}$ and two integer-based *design tiers*, $d_v$ and $d_t$, which control the complexity of the visual and textual inputs, respectively.

We use the color recognition task as an example. As shown in Figure 1(a), the visual design tier ($d_v$) modulates perceptual difficulty by adding distractors, shrinking the target object, or introducing occlusions. A low $d_v$ might feature a single, clear red square, while a high $d_v$ might present it as a small, partially obscured object among many other colorful shapes. Similarly, the textual design tier ($d_t$) controls reasoning complexity. A low $d_t$ provides a direct (but conflicting) statement (e.g., "The square is blue"), while a high $d_t$ requires multi-hop relational reasoning (e.g., "The square shares its color with a morpho butterfly's wings"). We ensure that the conflicting color mentioned in text never appears among visual distractors, so each modality provides information independently. By systematically pairing different levels of $d_v$ and $d_t$, we generate a structured landscape of conflict cases that spans a wide and predictable range of relative difficulty. Further details are in Appendix C.2.1.

### 3.2 QUANTIFYING PERCEIVED UNCERTAINTY WITH ENTROPY

**Entropy as proxy of perceived uncertainty.** While design tiers provide a human-interpretable notion of difficulty, our analysis requires a model-centric metric that reflects the model's *own* perceived uncertainty. For this purpose, we employ the **Entropy** of the model's output distribution over the answer token (Shannon, 1948; Cao & Ou, 2025). Given a unimodal input $x$ (either vision-only or text-only), for example, consider a vision-only input where the question is "What is the color of the square?" and the image shows a red square. Its uncertainty is:

$$H(x) = -\sum_{y \in \mathcal{V}} p(y \mid x) \log p(y \mid x),$$

where $\mathcal{V}$ is the token vocabulary. A low entropy value indicates a confident, sharp prediction (e.g., the probability for "red" is high, and near zero for other tokens), whereas a high entropy value would suggest that the model also considers alternative tokens (e.g., "orange," "brown"), revealing greater uncertainty about its own prediction. Since the output is always in the same token space, entropy serves as a unified and comparable measure of perceived uncertainty across both modalities, which we denote as $H^{(v)}$ for vision and $H^{(t)}$ for text.

**Analysis of Unimodal Entropy Trends.** To validate that entropy reliably captures our designed difficulty, we measured it across different models and tiers, with the results presented in Figure 2.

The data provides strong empirical support for our methodology through three key observations. First, entropy consistently increases with higher design tiers $(d_v, d_t)$, proving it aligns with our intended difficulty structure. This trend is especially clear in the vision modality, where for instance, the LLaVA-v1.6-7B model's entropy climbs steadily from approximately 0.25 at the lowest difficulty tier to over 1.5 at the highest. Second, the entropy values for both text and vision span a broad and comparable dynamic range from near-zero to over 1.75, which is crucial for creating conflict scenarios with diverse relative uncertainties. Third, and critically, the differences in entropy across models correspond to their known capabilities. The Qwen2.5-VL model, for example, consistently exhibits the lowest entropy, reflecting its strong performance, while we also observe expected scaling trends within model families, such as the LLaVA-v1.5-13B model showing generally lower visual uncertainty than its 7B counterpart.

> **Conclusion:** (1) We construct a novel dataset that allows for the systematic and independent control of reasoning difficulty across visual and textual modalities. (2) Output token entropy is a robust and reliable proxy for a model's perceived unimodal uncertainty, establishing it as a sound foundation for our analysis.

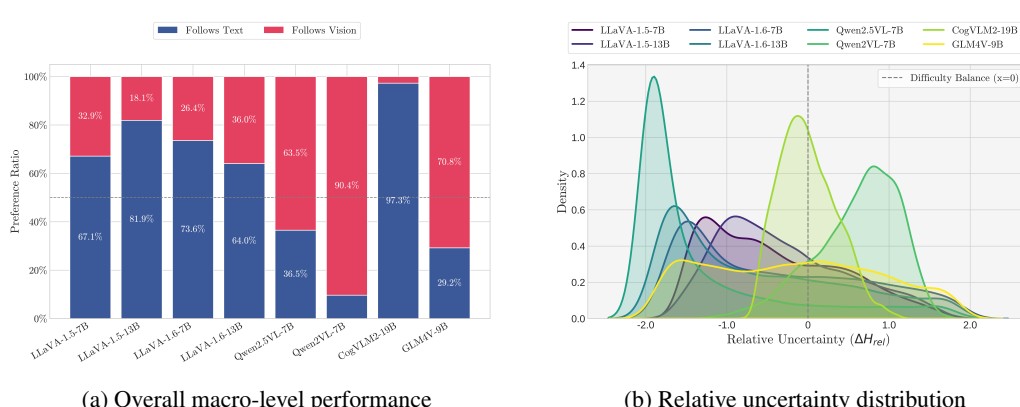

(a) Overall macro-level performance      (b) Relative uncertainty distribution

Figure 3: Macro-level modality-following ratios and relative uncertainty distributions of model performance on the dataset.

## 4    MODALITY FOLLOWING IS SHAPED BY RELATIVE UNCERTAINTY

**Contradictory Behaviors at the Macro Level.**    To thoroughly investigate modality following, we evaluated a diverse suite of MLLMs across three distinct datasets: our proposed toy dataset, the $MC^2$ benchmark (Zhang et al., 2025), and the modified $Pascal\ VOC$ dataset (Hua et al., 2025). Our model selection covers the LLaVA-1.5 (Liu et al., 2024a) and LLaVA-1.6 families (Li et al., 2024), the Qwen-VL series (Wang et al., 2024; Yang et al., 2024; Bai et al., 2025), as well as GLM-4V (GLM et al., 2024) and CogVLM2 (Hong et al., 2024). Detailed specifications of these models and the dataset construction process are provided in Appendix D and Appendix C.

As a first step, we analyzed the Text-Following Ratio (TFR) on instances where models correctly answer unimodal queries. Figure 3a presents the results on our Controlled Dataset, and the results reveal stark seemingly arbitrary differences between model families: the CogVLM2 and LLaVA series consistently exhibit a high TFR, appearing strongly text-following. In contrast, the GLM-4V and Qwen-VL series are notably more vision-following. This raises a puzzle: *why do models exhibit such divergent and seemingly fixed preferences when evaluated on the same dataset?* More macro-level statistics and relative uncertainty distributions for the other real-world datasets are provided in Appendix F.1.

**A Finer Lens: Relative Unimodal Uncertainty.**    The core flaw in macro-level statistics like TFR is that they ignore the model's case-by-case reasoning confidence. To capture this, we introduce **relative unimodal uncertainty** ($\Delta H_{\text{rel}}$). For a given conflicting input $x = (I, T, Q)$, we first decouple its components to measure the unimodal uncertainties. We calculate the text-only entropy,

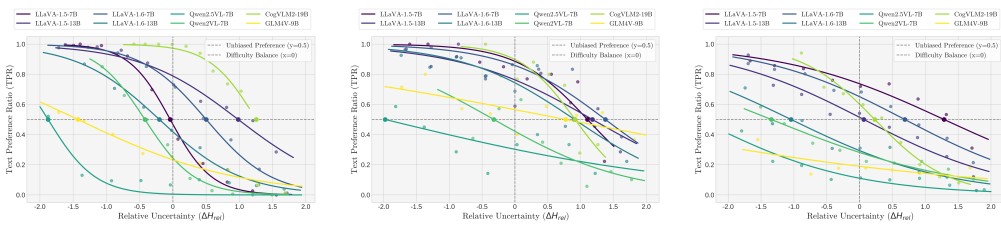

(a) Curve of our toy color recognition dataset.

(b) Curve of color recognition task in $MC^2$ dataset.

(c) Curve of object recognition task in $Pascal\ Voc$ dataset.

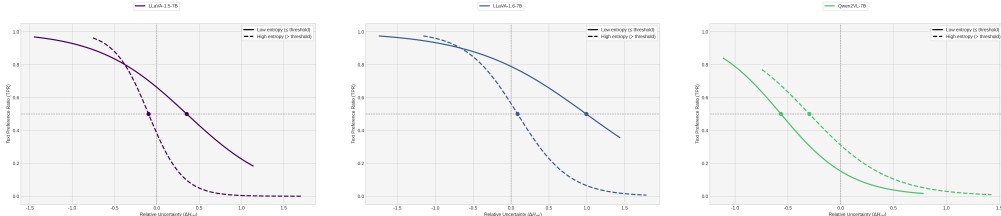

(d) Curves of LLaVA-1.5-7B model on our toy dataset.

(e) Curves of LLaVA-1.6-7B model on our toy dataset.

(f) Curves of Qwen2VL-7B model on our toy dataset.

Figure 4: **Universality and Robustness of the Relative Uncertainty Law across Datasets and Entropy Levels.** (a-c) General Monotonicity: The monotonic decrease of TPR is consistent across diverse datasets ; (d-f) Robustness to Absolute Uncertainty: The monotonic decrease of TPR is consistent across different absolute entropy levels

$H^{(t)}$, by providing only the text and the question $(T, Q)$ to the model. Similarly, we calculate the vision-only entropy, $H^{(v)}$, by providing only the image and the question $(I, Q)$. The relative uncertainty is the normalized difference between these two values:

$$\Delta H_{\text{rel}}(x) = \frac{2\left(H^{(t)}(x) - H^{(v)}(x)\right)}{H^{(t)}(x) + H^{(v)}(x)}.$$

Here, $H^{(t)}(x)$ and $H^{(v)}(x)$ refer to the unimodal entropies derived from the components of the multimodal input $x$. This metric, $\Delta H_{\text{rel}}$, thus quantifies the model's perceived confidence gap for each specific input. It is a direct manifestation of the model's **unimodal capabilities**, shaped by its architecture and training data. A negative value indicates the model is more confident in the text, while a positive value means it is more confident in the vision. When we plot the distribution of $\Delta H_{\text{rel}}$ for the correctly solved cases (Figure 3b), a new puzzle emerges. Despite their different macro-level behaviors, most models face a similar distribution skewed towards negative values, meaning the dataset is, on average, easier for them to process through text. This deepens the mystery: *if the underlying difficulty distribution is similar for most models, why are their final choices so different?*

**A Unified Monotonic Law.** The answer emerges when we shift our perspective from aggregate statistics to the dynamic relationship between uncertainty and choice. By plotting the probability of a model following the text modality against the corresponding $\Delta H_{\text{rel}}$ for each case, the apparent chaos resolves into a single, unified pattern, as shown in Figure 4a. For all eight models, regardless of architecture or scale, the curve shows a smooth, **monotonic decrease**. In other words, as text becomes harder relative to vision (i.e., as $\Delta H_{\text{rel}}$ increases), the probability that the model follows the text steadily and predictably decreases. This discovery directly confirms our central hypothesis from the Introduction: modality following is not a fixed trait but a dynamic behavior governed by relative reasoning uncertainty.

**Quantifying Inherent Preference via the Balance Point.** While all models obey this monotonic law, their curves are positioned differently along the axis. This leads to our second key insight. We define the **balance point** as the $\Delta H_{\text{rel}}$ value at which the model is equally likely to follow either

modality (a 50% text-following probability). This balance point provides a principled, quantitative measure of the model's **inherent modality preference**—the concept we introduced in the Introduction as the model's intrinsic leaning when reasoning difficulty is equalized. A balance point below zero indicates an inherent *vision preference* (as text must be significantly easier to be treated as equal), while a point above zero indicates an inherent *text preference*. This finally allows us to disentangle a model's fluid, in-the-moment decision-making from its stable, underlying biases.

**Reconciling Macro-Level Contradictions.** Our framework, which separates unimodal capability (reflected in the $\Delta H_{\text{rel}}$ distribution) from inherent preference (the balance point), can now fully explain the apparent contradictions from our initial macro-level analysis. Consider Qwen2-VL, which appears more vision-following than Qwen2.5-VL based on its VFR. Our analysis reveals this is largely a dataset artifact. Qwen2-VL's stronger visual capabilities on this specific dataset mean that more data points simply fall into the "vision-is-easier" (positive $\Delta H_{\text{rel}}$) region, mechanically inflating its vision-following stats. However, Qwen2.5-VL has a balance point further to the left (more negative), revealing a *stronger inherent vision preference*, as it continues to trust vision even when text is substantially easier. Similarly, the difference between LLaVA and Qwen models is not just about capability. While both face a dataset where text is often easier, Qwen models possess a clear inherent vision preference (negative balance point), whereas LLaVA models have a neutral or text-leaning preference (balance point near or above zero). It is this crucial difference in their *inherent preference* that drives their divergent behaviors, a nuance entirely missed by macro-level metrics.

**Generalization across datasets** we extended our evaluation to two real-world conflict benchmarks adapted from prior work: the $MC^2$ dataset (Zhang et al., 2025) and the conflict-modified $Pascal\ VOC$ dataset (Hua et al., 2025). As shown in Figure 4(a-c), despite the significant differences in image and textual description domains, the fundamental monotonic relationship between relative uncertainty ($\Delta H_{rel}$) and modality following remained consistent across all datasets. More results of other datasets are detailed in Appendix F.2.

**Robustness to absolute difficulty** To rule out artifacts of specific difficulty levels, we split the data into high-entropy (hard) and low-entropy (easy) subsets based on median total entropy. As illustrated in Figure 4(d-f), both subsets independently preserve the strict monotonic decline. Notably, high-entropy cases (dashed lines) exhibit steeper curves with balance points closer to zero compared to low-entropy cases (solid lines). This aligns with the intuition that uncertain models are more easily swayed by relative difficulty, whereas confident models show greater resistance. Additional comparisons are presented in Appendix F.3.

**Generalization across uncertainty metrics** To verify whether entropy is the unique prerequisite for our findings, we tested alternative metrics that share similar directionality in indicating uncertainty, specifically negative log-probability ($-\log p$) and prediction margin ($1-$margin). As detailed in Appendix F.4, the monotonic law holds robustly across these measures, confirming that the phenomenon is driven by the underlying uncertainty rather than the specific mathematical formulation. We reserve a deeper theoretical analysis of these metric choices for the Discussion section.

## 5 The Internal Mechanism: Oscillation in the Face of Ambiguity

Having established a robust behavioral law that modality following is a dynamic function of relative uncertainty, we now turn to the underlying mechanism. *Why* does a model hesitate and produce averaged following behavior when the relative uncertainty is close to its inherent balance point? In this section, we peer inside the model's layer-by-layer reasoning process to reveal the internal dynamics of its decision-making. Our analysis demonstrates that the model's external hesitation is a direct consequence of internal **oscillations** between the conflicting choices.

**Probing Layer-wise Predictions in Ambiguous vs. Clear Regions.** To quantify the internal dynamics of conflict resolution, we categorized scenarios into two distinct regions based on relative uncertainty: an *ambiguous region* ($\Delta H_{\text{rel}}$ near the balance point) and a *clear region* (one modality dominant). We tracked the top-1 prediction at each layer during forward using a *LogitLens*-style

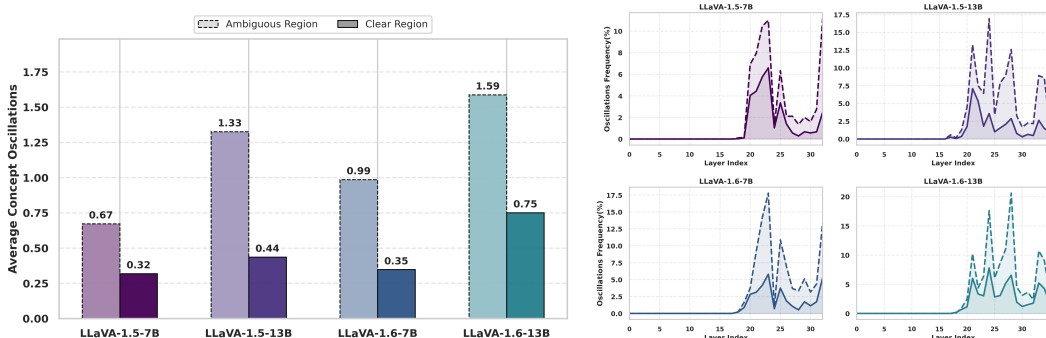

(a) Average Oscillations in ambiguous and clear region.     (b) Oscillations Frequency across model layers

Figure 5: Visualization of internal concept oscillations across different models. **(a) Average Oscillation Counts:** Display the average number of concept oscillations per sample. The lighter bars with dashed borders represent the **Ambiguous Region**, while the darker solid bars represent the **Clear Region**. **(b) Layer-wise Frequency Distribution:** Illustrate the oscillation frequency (%) across the model's layers (x-axis). **Dashed lines** correspond to the Ambiguous Region, and **solid lines** correspond to the Clear Region.

technique (nostalgebraist, 2020; Zhang et al., 2024b) and defined **oscillations** as switches between vision-supported and text-supported answers.

The comparative results are presented in Figure 5a. The bar charts reveal a stark contrast: the oscillation frequency in the ambiguous region is nearly double than that in the clear region across models. Figure 5b further illustrates the distribution of these oscillations across layers. We observe that for all models, oscillations predominantly initiate in the **middle-to-late stages** (typically after layer 15). Based on these observations, we derive three key conclusions:

(1) Models exhibit significantly higher instability when relative uncertainty is high in ambiguous regions. (2) This "hesitation" is not a global phenomenon but is strictly localized to the deeper layers of the network. (3) The intensified oscillation in ambiguous cases maps specifically to these later processing stages, suggesting that conflict resolution is a high-level cognitive process.

To validate the generality of these findings, we extended this analysis to all evaluated models across three additional datasets. As detailed in Appendix F.5, these extensive experiments demonstrate striking consistency in behavioral patterns, confirming that this layer-wise oscillation mechanism is a universal property of MLLMs when resolving multimodal ambiguity.

**Visualizing Indecision with Logit Difference Heatmaps.** To visualize this internal struggle, Figure 6a plots the logit difference (text-supported minus vision-supported logits) across layers for four representative models. Here, the x-axis denotes the layer index, and the y-axis represents relative uncertainty ($\Delta H_{\text{rel}}$). Crucially, the "white" neutral zones (indicating near-zero logit differences) do not strictly align with the geometric center ($y = 0$) but rather correspond to each model's unique **balance point** identified in Figure 4a. For instance, while LLaVA-1.5-7B and LLaVA-1.6-13B exhibit indecision near the center, the white bands for LLaVA-1.5-13B and LLaVA-1.6-7B shift noticeably towards the vision-easier region (positive $\Delta H_{\text{rel}}$). This confirms that internal hesitation is driven by the model's subjective equilibrium. Conversely, regions distant from this balance point exhibit early saturation to deep red or blue, signaling that once outside the ambiguous zone, models commit quickly and stably.

**A Case Study: The Dynamics of Conflict.** We exemplify these dynamics using a concrete case on LLaVA-1.5-7B (Figure 6b). By manipulating textual difficulty ($d_t = 0, 1, 2$) for a single image, we effectively shift the input across uncertainty regions. While easy ($d_t = 0$) and hard ($d_t = 2$) texts induce rapid, stable commitments to text (blue line) and vision (red line) respectively, the intermediate case ($d_t = 1$) falls into the ambiguous region. Here, the trajectory (gray line) vacillates near the zero-line decision boundary, visually capturing the internal hesitation driven by balanced uncertainty.

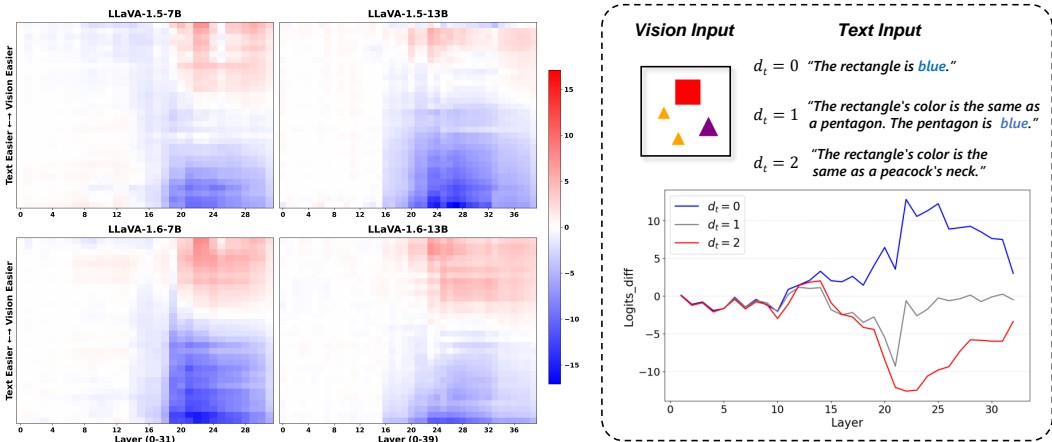

(a) Logit Difference Heatmap Across Model Layers and Relative Uncertainty.

(b) Case Study: Impact of Text Uncertainty on Layer-wise Confidence Dynamics.

Figure 6: Visualization of the Model's Internal Decision-Making Dynamics. In these visualizations, the x-axis represents the model's layers. The y-axis is the logit difference, calculated as the logits of the text answer minus the logits of the vision answer ($\text{logit}(Y_t) - \text{logit}(Y_v)$).

# 6 APPLICATION: GUIDING DATA SELECTION FOR PREFERENCE STEERING

**Motivation and Setup.** To explore how relative uncertainty guides preference steering, we conducted a controlled Supervised Fine-Tuning (SFT) experiment. We first split the dataset into training and testing sets, ensuring both contained samples across the full entropy spectrum. The training set was then used to construct four distinct subsets: three based on specific difficulty regions—**Pos** (Vision-Easier), **Neg** (Text-Easier), and **Mid** (Ambiguous)—and one **Uniform** set sampled globally. Models (LLaVA-1.5-7B and Qwen2-VL-7B) were fine-tuned to switch their modality preference (e.g., Vision → Text) and evaluated on the common stratified test set. Detailed dataset construction and training hyperparameters are provided in Appendix E.

**Results.** Figure 7a reports the target modality following rates on the test set after fine-tuning with training sets of varying relative uncertainty. Our observations are threefold: (1) **General Efficacy:** Compared to the baseline (blue line), all fine-tuned models (colored lines) show an improved propensity to follow the target modality across all regions. (2) **Consistency with Baseline Difficulty:** The post-tuning performance distribution mirrors the baseline's inherent difficulty. Even after finetuning, models consistently exhibit the lowest following rates in regions where the target modality was originally weakest ("hard" regions). For example, when steering towards Text (shown in panel a/c in Figure 7a), performance remains lowest in the Pos region (originally Vision-dominant). (3) **Inefficiency of "Easy" Data:** Training exclusively on data where the target modality is already dominant ("easy" data) proves to be the least efficient strategy for achieving general improvement. Our results reveals that models trained on these "easy" data struggle to generalize to complex scenarios. In the Vision-steering task (shown in panel b/d in Figure 7a), the **Pos-trained model** (orange line)—learned from easy, vision-dominant samples—consistently exhibits the poorest performance in the challenging Neg region, lagging behind both the Mid (gray) and Uniform (brown) models.

**Mechanism.** Figure 7b offers a mechanistic explanation for the inefficiency of "Easy" data by tracking post-tuning oscillation frequency changes (using Visual-steering as the primary example). First, we observe a distinct contrast between stabilization and struggle. In regions where the target modality is naturally easier or balanced (Pos and Mid), fine-tuning consistently yields a substantial reduction in oscillations (negative bars), signaling a confident commitment to the new preference. Conversely, in the **Hard (Neg) region**, oscillation frequencies show minimal reduction or even noticeable increases. This reflects a state of cognitive dissonance, where the model vacillates between its inherent prior and the newly tuned instruction.

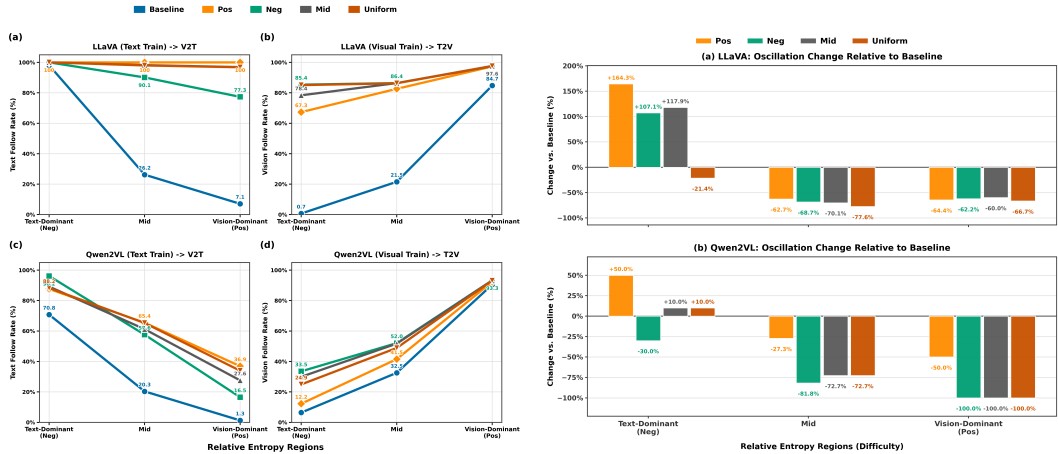

(a) Target Modality Follow Rate of Finetuned Models.  (b) Oscillation Change Ratio vs Baseline.

Figure 7: **Impact of Preference Finetuning and Internal Dynamics.** **(a)** The modality following probability curves under different data selection strategies (Pos, Neg, Mid, Uniform) across LLaVA15-7B and Qwen2VL-7B. **(b)** The relative change in internal "oscillation" (conflict) compared to the baseline. Negative values indicate a reduction in hesitation.

Crucially, this instability is exacerbated by "Easy" training. The **Pos-trained model** (orange bars)—which only saw easy visual dominance during training—exhibits the poorest oscillation control in the challenging Neg region. For instance, it triggers the most dramatic spike in oscillations (e.g., +164% for LLaVA), whereas models trained on harder data (e.g., Neg/Green) manage to better suppress this instability. This confirms that training on trivial samples fails to equip the model with robust conflict-resolution mechanisms, leaving it prone to high-frequency vacillation when facing genuine difficulty.

**Practical Implications.** Highlighting the failure of easy-to-hard generalization, our results suggest that data efficiency in preference alignment (e.g., SFT) hinges on reasoning uncertainty. Future strategies should therefore prioritize ambiguous and hard cases over trivial cases that merely reinforce existing priors.

# 7 DISCUSSION AND FUTURE WORK

While this work establishes a foundational law for conflict resolution in short-answer and discriminative tasks, we recognize that real-world multimodal interactions often involve complex, multi-step reasoning or open-ended generation (e.g., Chain-of-Thought reasoning). In such scenarios, conflicting information acts as a pivotal node within a longer reasoning chain rather than the sole output. A critical direction for future research is to investigate how relative uncertainty governs the branching of reasoning paths in these extended contexts. To capture the dynamics of such high-level cognition, future work must move beyond token-level entropy, employing advanced metrics like **semantic entropy** (Kuhn et al., 2023) or aggregated sequence probabilities to quantify relative uncertainty over entire concepts or reasoning steps.

# 8 CONCLUSION

We move beyond coarse dataset-level analysis by demonstrating that modality following is governed by relative reasoning uncertainty andinherent preference. We establish a fundamental law: the probability of following a modality decreases as uncertainty grows, with the **balance point** serving as a precise measure of preference. We further identify internal **oscillations** as the mechanism behind hesitation. This framework effectively disentangles capability from bias, providing crucial guidance for robust MLLM alignment.

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

## A  THE USE OF LARGE LANGUAGE MODELS (LLMS)

During the preparation of this paper, we used large language models (LLMs) solely as general-purpose writing assistants. Specifically, LLMs were employed to help refine the clarity, grammar, and readability of our drafts, as well as to suggest alternative phrasings in English. Importantly, all conceptual contributions including the design of research questions, development of methods, execution of experiments, and interpretation of results were conceived and carried out entirely by the authors. The authors carefully reviewed and edited all text suggested by LLMs to ensure accuracy and originality, and we take full responsibility for the final content of the paper.

## B  RELATED WORK

**Processing and Characterizing Conflicting Information.**   A significant body of research has focused on characterizing how Multimodal Large Language Models (MLLMs) behave when faced with conflicting inputs. Various benchmarks have been developed to probe this phenomenon, revealing a complex and often inconsistent landscape of modality preferences. A frequently reported observation is that many models exhibit a "blind faith" in text, systematically ignoring visual information Deng et al. (2025). However, this tendency is not universal, as other studies demonstrate that preferences can vary significantly across different models and scenarios Zhang et al. (2025); Liu et al. (2024b). Further work with benchmarks like MMIR has focused on the model's ability to detect and reason about such inconsistencies (Yan et al., 2025). The lack of a consistent principle to explain these varied and often contradictory macro-level observations is a key motivation for our work. Our primary contribution is to move beyond dataset-level statistics by proposing a unifying framework. We explain this apparent variability as an emergent property of two core factors: case-specific **relative reasoning uncertainty** and a model's stable **inherent preference**.

**Explaining and Interpreting Conflict Resolution.**   Another line of research seeks to explain the underlying causes of modality preference. Some studies focus on external factors that can steer a model's behavior, such as the order of inputs (Deng et al., 2025) or the use of instructional prompts. Others delve deeper, attributing the behavior to internal factors like inconsistencies within the model's learned knowledge representations Zhu et al. (2024); Golovanevsky et al. (2025). A third approach uses attribution methods, such as those based on Shapley values, to quantify the relative influence of each modality on the final decision (Alishahi et al., 2019; Parcalabescu & Frank, 2022; 2024). While these approaches identify potential causes and influencing factors, they do not fully reveal the dynamic, layer-by-layer computational process through which a model resolves ambiguity. Motivated by this gap, our work provides this missing mechanistic link. We introduce the concept of internal "oscillations" as direct, observable evidence of the conflict resolution process, demonstrating how our high-level framework is physically manifested in the model's computational dynamics and explains *why* models hesitate under uncertainty.

**Modality Reliance and Pre-training Priors**   Existing research has extensively probed how multimodal models utilize and integrate different modalities, often revealing a tendency toward modality imbalance. (Gat et al., 2021) introduced the **Perceptual Score**, a permutation-based metric to quantify a model's reliance on specific modalities, revealing that models frequently exploit textual shortcuts while ignoring visual inputs. Similarly, (Frank et al., 2021) utilized cross-modal input ablation to diagnose information flow, uncovering a significant asymmetry: while models effectively recruit vision for language tasks ("Vision-for-Language"), they often fail to leverage textual input for visual recognition ("Language-for-Vision"). The origins of this textual dominance are further elucidated by (Han et al., 2025), who demonstrate that Large Language Models (LLMs) acquire rich "visual priors" and reasoning capabilities solely from text pre-training. This suggests that the strong inherent preference for text observed in MLLMs is rooted in these latent, pre-existing capabilities developed before visual alignment.

While previous studies characterize static modality dependence or identify the pre-training origins of global biases, they do not fully explain the dynamic decision-making process when modalities actively contradict one another during inference. Our work advances this line of inquiry by shifting focus from static reliance to post-alignment inference dynamics. We propose that modality following is not a fixed attribute but a fluid behavior governed by Relative Reasoning Uncertainty. In this

framework, the "inherent preference" shaped by pre-training priors (as discussed (Han et al., 2025)) serves as a baseline balance point, which is dynamically modulated by the case-specific confidence gap between unimodal reasoning paths. Furthermore, we uncover the internal mechanism of this conflict resolution through layer-wise oscillations, bridging the gap between the static capabilities identified in prior work and the dynamic hesitation observed during runtime.

# C DATASET CONSTRUCTION

## C.1 REAL-WORLD INFORMATION CONFLICT DATASETS

To validate the generalization of our findings across broader and more realistic tasks, we adapted two real-world information conflict datasets—$MC^2$ (Zhang et al., 2025) and the synthetic dataset used by (Hua et al., 2025) into formats compatible with our methodology.

$MC^2$ is a benchmark dataset designed to evaluate the modality preferences of multimodal large language models (MLLMs). It consists of 2,000 samples sourced from TDIUC(Kafle & Kanan, 2017), covering eight types of perceptual tasks such as color detection, counting, and object recognition. The dataset was constructed through a semi-automated process:

- Images, questions, and corresponding visual answers were first collected from TDIUC;
- Large language models were employed to generate conflicting textual contexts and distracting answers that contradict the visual content;
- Baseline MLLMs were used to filter samples that could be correctly understood via a single modality;
- Multiple rounds of human validation were conducted to ensure genuine inter-modal conflicts and distinct answers supported by each modality.

Each case includes an image, a conflict text context, a question, and two modality-specific answers (text-based and vision-based). For our study, we selected cases from the color detection and object recognition tasks and further filtered them to include only those with single-word answers, so as to avoid potential ambiguities in answer evaluation.

The other real-world information conflict dataset we used was adapted from (Hua et al., 2025), which synthesized a dataset based on Pascal VOC (Everingham et al., 2010) using an adversarial sampling approach. The core methodology involves:

- Selecting a real image from the original dataset;
- Randomly sampling an incorrect class label that differs from the ground-truth label;
- Using a standardized template to generate a caption that contradicts the image content;
- Constructing multiple-choice options to systematically create evaluation samples with inconsistent image-text information.

Following this sampling and synthesis procedure, we constructed 2,470 multiple-choice samples from the Pascal VOC train subset. Unlike the original study, which employed templates with explicit image or text-biased instructions, we used several neutral question templates to ensure fair treatment of both textual and visual information. Table 1 presents the templates used in our dataset construction.

## C.2 SYNTHETIC-SCENARIO INFORMATION CONFLICT DATASET

To investigate the external performance and internal mechanisms of multimodal models when dealing with conflicts between image and text information, we constructed two Synthetic-scenario information conflict datasets. The first is **Color Recognition Dataset**, which requires the model to identify the color of geometric shapes placed on a white canvas. The second is **Attribute Recognition Dataset**, adapted and filtered from the CLEVR(Johnson et al., 2017) dataset, whose task is to identify the material and shape of three-dimensional objects. Both datasets contain multiple task groups. Each group provides images with increasing visual complexity and text descriptions that

Table 1: Templates for Modified Pascal Dataset

| **Caption templates** |
| --- |
| This is an image of a {CLASS_LABEL}. |
| This is a photo of a {CLASS_LABEL}. |
| An image of a {CLASS_LABEL}. |
| A photo of a {CLASS_LABEL}. |
| This is a {CLASS_LABEL}. |
| A {CLASS_LABEL}. |
| **Question templates** |
| What is being shown? |
| What is presented? |
| What is depicted? |
| What is this? |
| **Option template** |
| Select from the following classes: |
| **Instruction template** |
| Answer the question using a single word or phrase. |
| **Answer template** |
| Image_Answer: |
| Text_Answer: |

contradict the image information while exhibiting increasing textual reasoning complexity. By systematically controlling the visual perception complexity ($d_v$) and the textual reasoning complexity ($d_t$), this design constructs conflict scenarios with diverse visual-textual difficulty combinations in a systematic manner.

### C.2.1 DATASET OVERVIEW

The Color Recognition Dataset consists of 400 groups, each containing 14 images and questions with 3 different types of conflict descriptions. Images with difficulty levels 0–4 are 800×600 pixels, while those with levels 5–13 are 224×224 pixels. The text is divided into three different types, with an average length of 22.7 words. In each group, the same image_answer color can be derived from any image information, while the same text_answer color which is different from the image_answer, can be obtained from any conflict description in the text. The distribution of image_answer and text_answer is as follows:

- **Image_answer Colors:** Red(67), Yellow(67), Blue(67), Green(66), Purple(66), Orange(67)
- **Text_answer Colors:** Red(67), Yellow(66), Blue(67), Green(67), Purple(66), Orange(67)

The Shape subset and the Material subset of the Attribute Recognition Dataset each contain 300 groups. Each group includes 4 images and questions with 3 different types of conflict descriptions. All images are 480×320 pixels, while the text is divided into five different types, with an average length of 30.0 words. In each group, the same image_answer attribute can be derived from any image information, while the same text_answer attribute which is different from the image_answer can be obtained from any conflict description in the text. The distribution of image_answer and text_answer is as follows:

- **Image_answer Shapes:** Sphere(108), Cube(100), Cylinder(92)
- **Text_answer Shapes:** Sphere(100), Cube(92), Cylinder(108)
- **Image_answer Materials:** Metal(160), Rubber(140)
- **Text_answer Materials:** Metal(140), Rubber(160)

### C.2.2 IMAGE GENERATION OF COLOR RECOGNITION DATASET

For each set of 14 images with a progressive difficulty gradient in the Color Recognition Dataset, we used the Python PIL library for rendering. The following is the generation pipeline.

1. **Initialization:** A **target shape** (e.g., Circle) is randomly selected.

2. **Color Assignment:**
   - **Visual Answer Color:** One color is randomly assigned to the target shape.
   - **Textual Answer Color:** A different color is randomly selected as the conflicting textual statement.

3. **Distractor Generation:** Distractor shapes are randomly chosen from the set excluding the target shape. Their colors are randomly selected from the set excluding both the visual and textual answer colors.

4. **Difficulty Tiers ($d_v = 0$ to 13):** Fourteen progressive difficulty levels are defined by target size, number of distractors and occlusion. Parameters are specified in Table2.

Table 2: Visual Difficulty ($d_v$) Tiers Specification

| Difficulty($d_v$) | Target Size | # Distractors | Occlusion Rule |
|:---:|:---:|:---:|:---:|
| 0 | 80-200 pixels | 0 | No occlusion |
| 1 | 80-200 pixels | 1 | No occlusion |
| 2 | 80-200 pixels | 2 | No occlusion |
| 3 | 80-200 pixels | 3 | No occlusion |
| 4 | 80-200 pixels | 4 | No occlusion |
| 5 | 20%-40% of image | 7 | 50% occlusion rate |
| 6 | 20%-40% of image | 10 | 80% occlusion rate |
| 7 | 5%-10% of image | 7 | 50% occlusion rate |
| 8 | 5%-10% of image | 11 | 80% occlusion rate |
| 9 | 4%-6% of image | 20 | 30% occlusion rate |
| 10 | 4%-6% of image | 30 | 60% occlusion rate |
| 11 | 4%-6% of image | 40 | 50% occlusion rate |
| 12 | 4%-6% of image | 55 | 60% occlusion rate |
| 13 | 4%-6% of image | 70 | 70% occlusion rate |

**Note 1:** "Occlusion rate" refers to the proportion of distractors that visually overlap the target. Different rates for odd/even tiers introduce finer-grained difficulty variation.

### C.2.3 IMAGE SELECTION OF ATTRIBUTE RECOGNITION DATASET

All images in the Attribute Recognition Dataset were curated from the CLEVR dataset, which contains objects defined by three geometric shapes (cube, sphere, cylinder), two materials (rubber, metal), and eight colors. For each target attribute corresponding to the subset, our selection procedure began by forming all possible attribute–color pairs via the Cartesian product. For each unique pair, we identified images from the CLEVR validation set containing *exactly one* object matching that specific combination. The selected images were then assigned a difficulty level based on scene complexity, with a fixed number of images sampled per level to construct the final task groups.Table3 shows the various difficulty levels of the pictures.

Table 3: Difficulty levels for image selection

| Difficulty($d_v$) | Number of objects in scene | Target object size |
|:---|:---|:---|
| 0 | 3–4 objects | large |
| 1 | 6–8 objects | large |
| 2 | 6–8 objects | small |
| 3 | ≥10 objects | small |

### C.2.4 TEXTUAL MODALITY CONSTRUCTION

The conflict text issues between the Color Recognition Dataset and the Attribute Recognition Dataset share many similarities in terms of structure and pipeline construction. In both cases, we gradually increase the complexity of the textual modality by increasing the number of reasoning steps and converting explicit reasoning into implicit reasoning. The questions within the same group share a fixed **target_shape** with the images of that group, inquire an **attribute** depending on the dataset they belong to, and utilize an identical **text_answer** that contradicts the image information. Each textual problem follows the format of: [Conflict Description] + [Question] + [Command].

- **Question:** `What {attribute} is the {target_shape}?`

- **Command:** `Please use one word to answer this question.`

For each group, we generate 3 types of conflict description for Color Recognition Dataset and 4 for Attribute Recognition Dataset with increasing difficulty. The Table4 below lists each type and a concise description, where **A** denotes the target_object, **T** denotes the text_answer, **B/S1/S2** represent randomly selected objects absent from the image, **D** represents a real-world instance unambiguously possessing attribute T, and **Pos1/Pos2** denote a pair of opposite spatial relations Left and Right.

Table 4: Question types and descriptions (descriptions only)

| Difficulty($d_t$) | Type | Description |
| --- | --- | --- |
| x | Original | No interference description. |
| 0 | Direct | The A is T. |
| 1 | Indirect_simple | The A's {attribute} is the same as a B. The B is T. |
| 2 | Indirect | The A's {attribute} is the same as a D. |
| 3 | Space(Attribute Recognition Dataset only) | There is a T S1, on the Pos1 of the S1 is a S2. The A's {attribute} is the same as the object Pos2 to the S2. |

**Robustness Processing:** To prevent models from solving tasks via superficial pattern matching, texts in Color Recognition Dataset for $d_t \geq 0$ were paraphrased using Qwen-Plus(Alibaba Cloud / QwenLM, 2025). This process preserved core semantics, reasoning structure, and key information tokens while varying sentence structure, prepositional phrases, and lexical choices.

**Control Group Setup:** For ablation studies, two types of control data were constructed:

- **Text-Irrelevant:** The target shape 'A' in conflict description only is replaced with a randomly chosen **non-target shape** (e.g., if target is 'circle', replace with 'triangle' or 'rectangle').

- **Image-Irrelevant:** The target shape 'A' in the entire text is replaced with a shape **never present** in the images ('star', 'cone', 'frustum'), maintaining the correspondence between the question and the text description while severing the connection with the image.

---

**Rewrite Questions Task**

====SYSTEM====
You are a conservative paraphrasing assistant specialized in subtle wording changes. Your goal is to rewrite a single question sentence while preserving *all* facts, *all* explicit instructions, and the exact multi-hop reasoning structure (number of inference steps and intermediate referents). Make only minor wording, grammar, punctuation, and token-count adjustments; do NOT add, remove, or transform factual content or the logical chain.

====USER====

Field type:
{FIELD_TYPE}
Original question:
{ORIGINAL_QUESTION}
Rewrite Instructions (STRICT):
1. Output exactly one rewritten question sentence (no explanation, no notes, no extra punctuation before/after).
2. Preserve *all* factual propositions and named referents. Do not add or remove facts.
3. Preserve the multi-hop reasoning structure:
- If the original is a single-step (direct), keep it single-step.
- If it is implicit multi-step (indirect), keep it implicit and do not make steps explicit.
- If it is explicit multi-hop (indirect_simple), keep the same explicit chain of premises and the same number of hops.
4. Preserve any explicit answering instruction exactly (e.g., "Please use one word to answer this question.").
5. Do not change the identity of entities (e.g., "hexagon", "pine tree", "circle") or the target attribute (e.g., "color").
6. Only rewrite wording, punctuation, and sentence flow to be more natural or shorter, and optionally reduce/increase token count slightly. You can use near-synonyms with very high similarity.
7. Avoid introducing pronouns that obscure referents; keep clarity of which object each premise references.
8. If the original contains multiple sentences that together form the multi-hop chain, you may combine or split them only if you exactly preserve the same premises and hop order.
Output: the single rewritten question sentence (no extra text).

## C.3 ILLUSTRATIVE SAMPLES FROM THE DATASET

To provide a more intuitive understanding of our image-text conflict dataset, we have sampled several image-question pairs from the Color Recognition Dataset, the Attribute Recognition Dataset subsets, MC$^2$ and modified Pascal dataset, and presented them in Figure8, Figure9, Figure10, Figure11 and Figure12.

**MC2 195894**

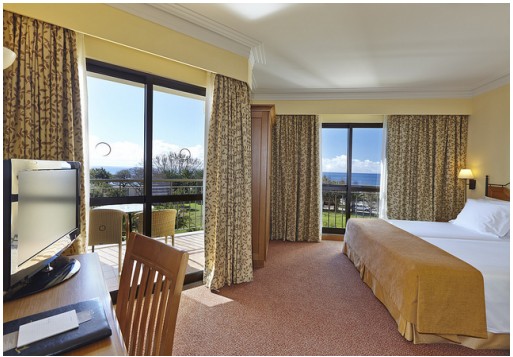

**Caption:** The bedroom ... The sheets on the large bed are made of a deep blue fabric ...

**Question:** What color is the sheet?
**Command:** Please use one English word to answer this question.
**Vision-based Answer:** *white*
**Text-based Answer:** *blue*

**MC2 518363**

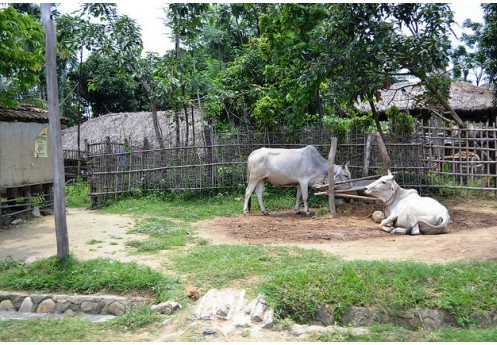

**Caption:** Two eagles perched on a fence in a backyard, with ...

**Question:** What animal is shown in the photo?
**Command:** Please use one English word to answer this question.
**Vision-based Answer:** *cow*
**Text-based Answer:** *eagle*

Figure 8: A selection of image-text pairings from MC$^2$ Dataset. The text highlighted in red indicates the descriptions and answers that conflict with the image information.

**Pascal Group730**

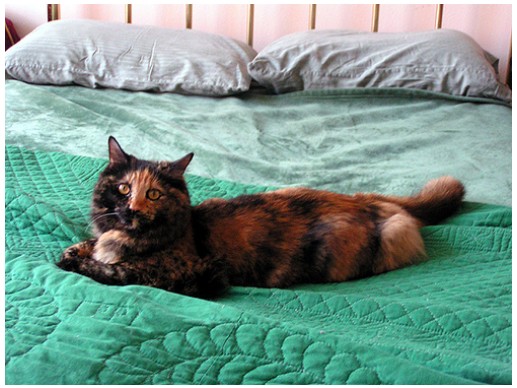

**Caption:** An image of a dog.
**Selections:** Select from the following classes: cow, potted plant, dog, bicycle, cat.
**Question:** What is this?
**Command:** Answer the question using a single word or phrase.
**Vision-based Answer:** *cat*
**Text-based Answer:** *dog*

**Pascal Group1453**

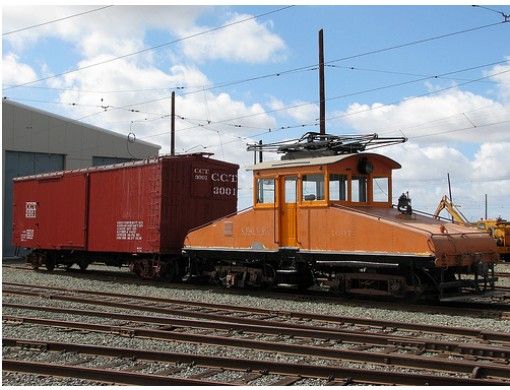

**Caption:** A photo of a bird.
**Selections:** Select from the following classes: sheep, train, bird, sofa, potted plant.
**Question:** What is being shown?
**Command:** Answer the question using a single word or phrase.
**Vision-based Answer:** *train*
**Text-based Answer:** *bird*

Figure 9: A selection of image-text pairings from modified Pascal VOC Dataset. The text highlighted in red indicates the descriptions and answers that conflict with the image information.

**Group41 Difficulty0**                    **Group41 Difficulty3**

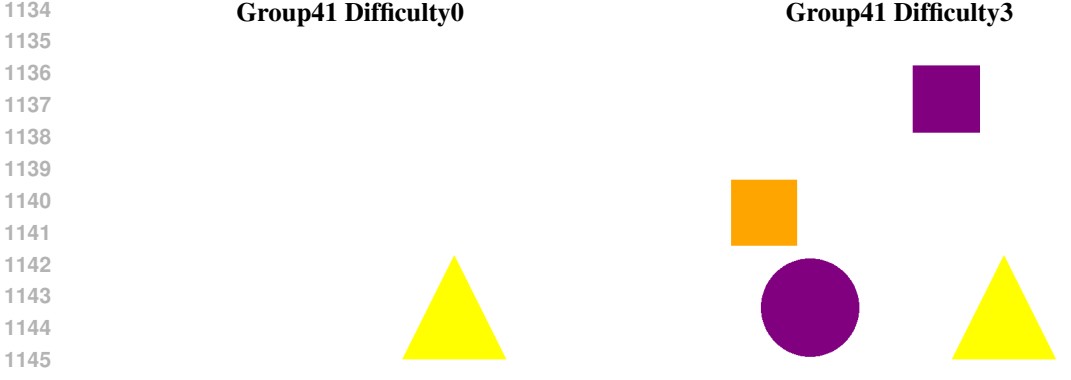

**Original:**
**Question:** What color is the triangle?
**Command:** Please use one word to answer this question.
**Vision-based Answer:** *Yellow*
**Text-based Answer:**

**Direct:** The triangle is blue.
**Question:** What color is the triangle?
**Command:** Please use one word to answer this question.
**Vision-based Answer:** *Yellow*
**Text-based Answer:** *Blue*

**Group41 Difficulty6**                    **Group41 Difficulty15**

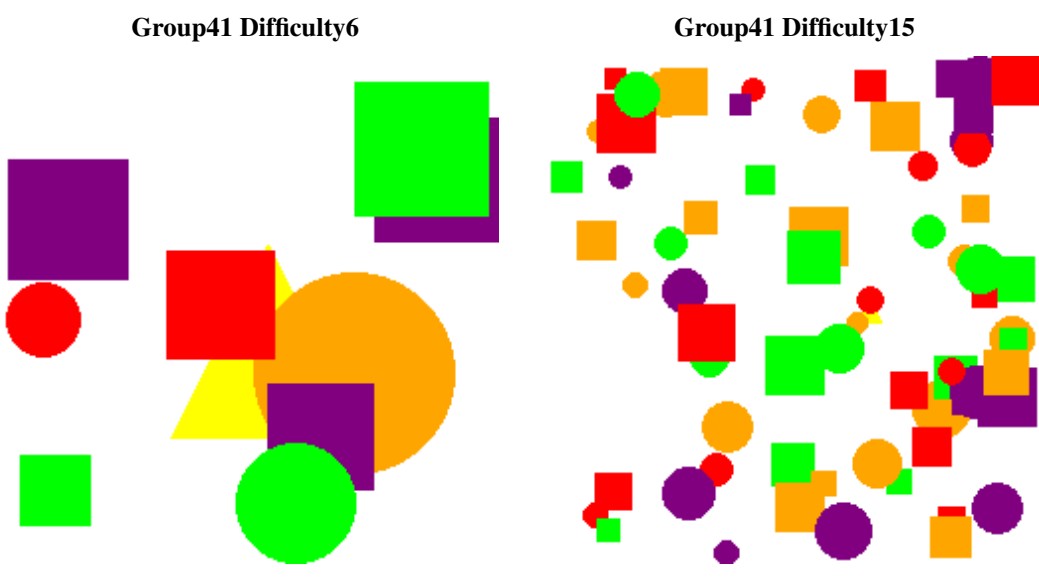

**Indirect_simple:** The triangle's color is the same as a pentagon. The pentagon is blue.
**Question:** What color is the triangle?
**Command:** Please use one word to answer this question.
**Vision-based Answer:** *Yellow*
**Text-based Answer:** *Blue*

**Indirect:** The triangle's color is the same as a mailbox in the US.
**Question:** What color is the triangle?
**Command:** Please use one word to answer this question.
**Vision-based Answer:** *Yellow*
**Text-based Answer:** *Blue*

Figure 10: A selection of image-text pairings from a group in the Color Recognition Dataset. The text highlighted in red indicates the descriptions and answers that conflict with the image information.

**Group193 Difficulty0**

**Group193 Difficulty2**

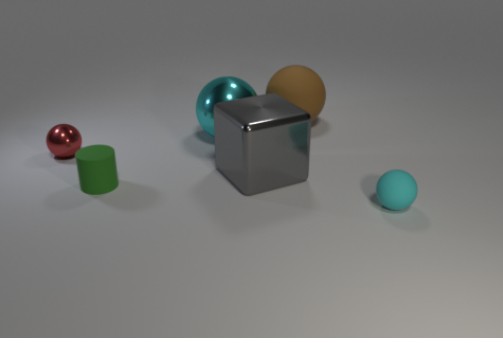

**Direct:** The cyan rubber object is a cylinder.

**Question:** What is the shape of the cyan rubber object?
**Command:** Please answer with one word.
**Vision-based Answer:** *sphere*
**Text-based Answer:** *cylinder*

**Indirect:** The cyan rubber object's shape is the same as a log.
**Question:** What is the shape of the cyan rubber object?
**Command:** Please answer with one word.
**Vision-based Answer:** *sphere*
**Text-based Answer:** *cylinder*

Figure 11: A selection of image-text pairings from a group in the Shape subset of the Attribute Recognition Dataset. The text highlighted in red indicates the descriptions and answers that conflict with the image information.

**Group79 Difficulty1**

**Group79 Difficulty3**

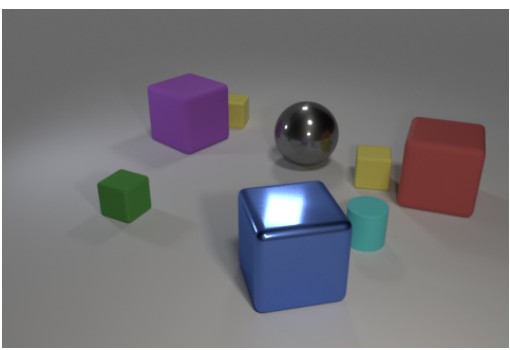
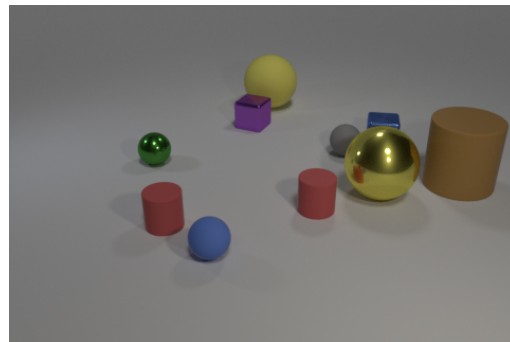

**Indirect_simple:** The Frustum is rubber, blue cube's material is the same as the Frustum.

**Question:** What is the material of the blue cube?
**Command:** Please use one word to answer this question.
**Vision-based Answer:** *metal*
**Text-based Answer:** *rubber*

**Space:** There is a rubber cone, the right of the cone is a wood frustum. The blue cube's material is the same as the object left to the wood frustum.
**Question:** What is the material of the blue cube?
**Command:** Please use one word to answer this question.
**Vision-based Answer:** *metal*
**Text-based Answer:** *rubber*

Figure 12: A selection of image-text pairings from a group in the Material subset of the Attribute Recognition Dataset. The text highlighted in red indicates the descriptions and answers that conflict with the image information.

## D DETAILS OF EVALUATED MODELS

To ensure a comprehensive evaluation and verify the generalization capabilities of our findings, we selected four representative Vision-Language Model (VLM) families. These models were chosen

to encapsulate diverse architectural strategies in visual encoding and modality alignment.Table 5 illustrates the key structural differences among these models.

Table 5: Overview of VLM families included in our experiments. *Deep Fusion* refers to modality interaction within LLM layers.

| Model | LLM Backbone | Visual Encoder | Resolution Strategy | Fusion Type |
|---|---|---|---|---|
| LLaVA-1.5 | Llama | CLIP-ViT-L | Fixed | MLP Projection |
| LLaVA-1.6 | Vicuna | CLIP-ViT-L | AnyRes | MLP Projection |
| Qwen2.5-VL | Qwen2.5 | SigLIP-based | Native Dynamic | MLP + M-RoPE |
| GLM-4v | GLM-4 | EVA-CLIP | High-Res Fixed | MLP Projection |
| CogVLM2 | Llama-3 | EVA-CLIP | High-Res Fixed | Deep Fusion |

# E  APPENDIX: PREFERENCE STEERING EXPERIMENT DETAILS

## E.1  DATASET CONSTRUCTION

To account for the unique uncertainty distribution of each model, we computed $\Delta H_{\rm rel}$ using the respective base models (LLaVA-1.5-7B and Qwen2-VL-7B) individually. We utilized a consistent held-out testing set of **8,409 samples** for evaluation.

For the training data, we partitioned the samples into subsets based on a threshold of $\delta = 0.5$:

- **Pos (Vision-Easier):** $\Delta H_{\rm rel} > 0.5$.
- **Neg (Text-Easier):** $\Delta H_{\rm rel} < -0.5$.
- **Mid (Ambiguous):** $|\Delta H_{\rm rel}| \leq 0.5$.
- **Uniform:** A random selection stratified to cover the full entropy spectrum.

To isolate the effect of data difficulty from data scale, we downsampled all training subsets to a unified size for each model: **1,614 samples** per subset for LLaVA-1.5 and **782 samples** per subset for Qwen2-VL.

## E.2  TRAINING SETUP

We performed Supervised Fine-Tuning (SFT) using LoRA with a rank of $r = 32$ and $\alpha = 32$. Models were trained for 1.5 epochs to ensure convergence without overfitting. The specific configurations were:

- **Qwen2-VL-7B:** Learning rate of $2e{-}5$. Target modules included attention layers: `["q_proj", "k_proj", "v_proj", "o_proj"]`.
- **LLaVA-1.5-7B:** Learning rate of $3e{-}5$. Target modules included both attention and MLP layers: `["q_proj", "k_proj", "v_proj", "o_proj", "gate_proj", "up_proj", "down_proj"]`.

# F  MORE EXPERIMENTAL ON VARIOUS MODELS AND DATASETS

## F.1  MACRO-LEVEL PERFORMANCE AND RELATIVE UNCERTAINTY DISTRIBUTION MEASURE

To evaluate the universality of our framework and findings, we extend our analysis to three additional task settings beyond those discussed in the main text: Color Recognition and Object Recognition on the $MC^2$ dataset, as well as Object Recognition on the Pascal VOC dataset.As detailed in Figure 13, we present:

- Macro-level Performance (Left Column): A summary of the aggregate modality-following behaviors (Text-Dominant vs. Vision-Dominant), illustrating the global preference tendencies of different models on these specific tasks.

- Relative Uncertainty Distribution (Right Column): The density plots of the relative reasoning uncertainty ($\Delta H_{\text{rel}}$).

Consistently, we observe that the distribution of uncertainty varies significantly across tasks and models, further necessitating the need for uncertainty-aware analysis.

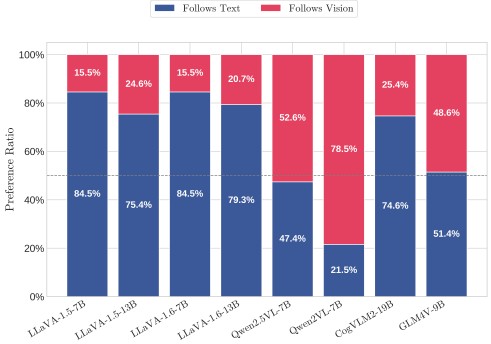
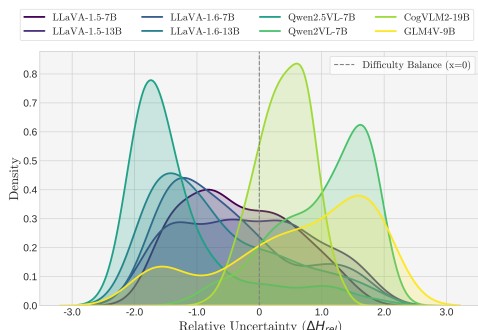

(a) Overall macro-level performance of color recognition task in $MC^2$ dataset

(b) Relative uncertainty distribution of color recognition task in $MC^2$ dataset

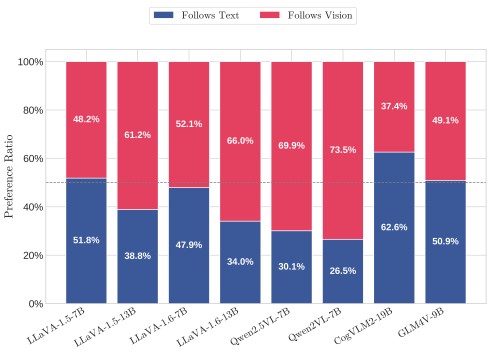
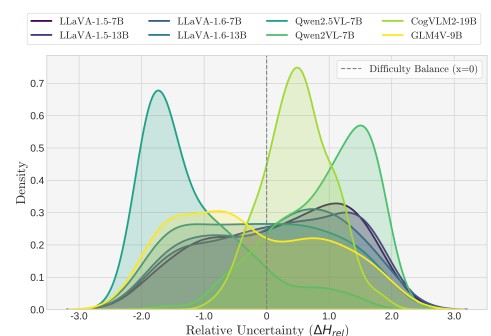

(c) Overall macro-level performance of object recognition task in $MC^2$ dataset

(d) Relative uncertainty distribution of object recognition task in $MC^2$ dataset

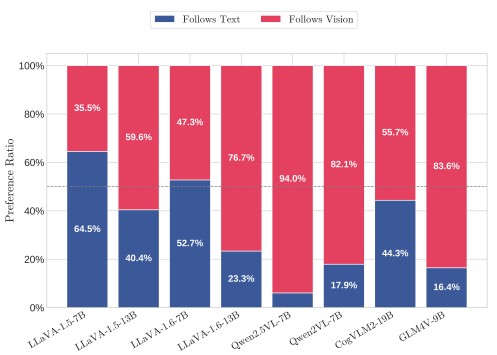
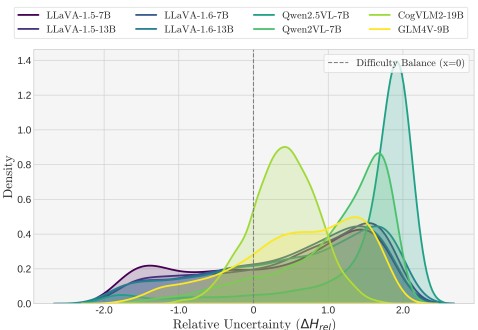

(e) Overall macro-level performance of object recognition task in $Pascal\ VOC$ dataset

(f) Relative uncertainty distribution of object recognition task in $Pascal\ VOC$ dataset

Figure 13: Macro-level modality-following ratios and relative uncertainty distributions of model performance on the dataset.

## F.2 CONSISTENCY OF MONOTONICITY: GENERALIZATION ACROSS DATASETS AND SETTINGS

To further substantiate the universality of the monotonic relationship between relative uncertainty and modality following, we extend our evaluation beyond the primary datasets discussed in the main text. As illustrated in Figure 14, we analyze three distinct scenarios to test the boundaries of our findings:

- **Generalization to Complex Tasks ($MC^2$ Object Recognition):** Unlike simple color recognition, object recognition requires semantic understanding of shape and texture. Figure 14a confirms that the monotonic law remains valid even for these semantic-heavy tasks.
- **Generalization to Attribute Binding (Attribute Recognition):** Using a subset from our Controlled Dataset focused on attribute binding (e.g., "the shiny metallic cube"), Figure 14b shows that the law governs not just global perception but also fine-grained feature grounding.
- **Robustness to Prompt Phrasing (Prompt Rewriting):** To rule out the possibility that the model's preference is an artifact of specific sentence structures, we diversified the prompts of the Color Recognition task using Qwen-rewrites (as detailed in Appendix C.2.4). Figure 14c demonstrates that the monotonic trend is invariant to linguistic variations.

Across all these varied dimensions—task type, reasoning complexity, and linguistic formulation—the fundamental law holds: models become less likely to follow a modality as their relative uncertainty regarding that modality increases.

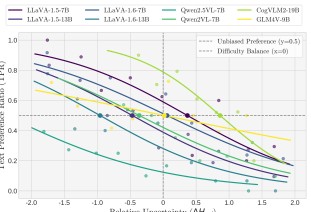 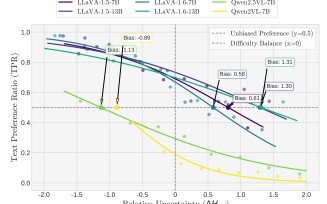 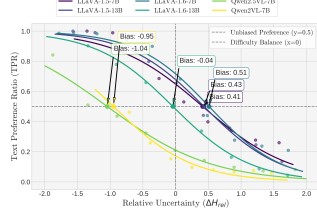

(a) $MC^2$ Dataset: **Object Recognition**

(b) Controlled Dataset: **Attribution Recognition**

(c) Robustness Check: **Prompt Diversification**

Figure 14: **Generalization of Modality Following Curves across Tasks and Settings.** We validate the monotonic law on: **(a)** Real-world Object Recognition tasks from the $MC^2$ benchmark; **(b)** Attribute Binding tasks from our Controlled Dataset; and **(c)** Color Recognition tasks with diversified prompts (rewritten by Qwen) to test linguistic robustness. Consistent with our main findings, the Text Following Ratio (TFR) monotonically decreases as relative reasoning uncertainty ($\Delta H_{rel}$) increases across all settings. Note that the shift in balance points reflects the models' inherent preferences varying with task characteristics.

## F.3 CONSISTENCY OF MONOTONICITY: HIGH VS. LOW ABSOLUTE UNCERTAINTY

We extend our analysis of absolute difficulty to three additional large-scale models: **LLaVA-1.5-13B**, **LLaVA-1.6-13B**, and **Qwen2.5-VL-7B**. Using the same methodology as the main text, we partition the Controlled Dataset into high-entropy (dashed lines) and low-entropy (solid lines) subsets based on the median total entropy.

As shown in Figure 15, two key observations emerge:

- **Universal Monotonicity:** Consistent with our fundamental law, all three models independently preserve the strict monotonic decline of modality following probability as relative reasoning uncertainty increases, regardless of the absolute entropy level.
- **Impact of Absolute Difficulty:**
    - For **LLaVA-1.5-13B** and **Qwen2.5-VL**, the results align with our main findings: high-entropy (hard) cases exhibit steeper curves with balance points shifting closer to zero

(the center). This confirms that high absolute uncertainty typically makes models more susceptible to relative cues.

– **LLaVA-1.6-13B** presents a notable exception. While it adheres to the monotonicity law, its high-entropy curve does not significantly shift toward the center compared to the low-entropy curve. This suggests that LLaVA-1.6-13B possesses a more rigid internal calibration, maintaining stable preference dynamics even under high overall uncertainty.

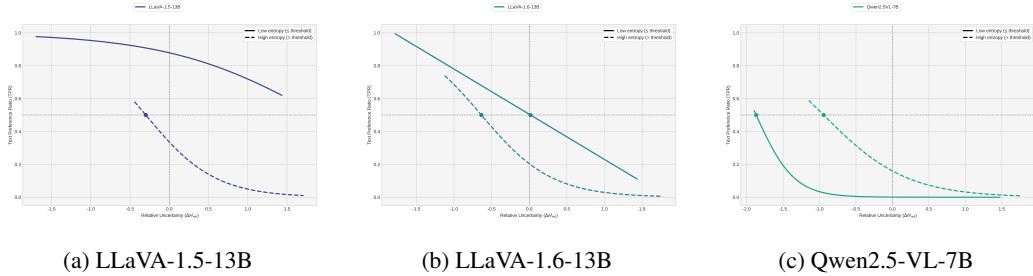

(a) LLaVA-1.5-13B      (b) LLaVA-1.6-13B      (c) Qwen2.5-VL-7B

Figure 15: **Robustness to Absolute Uncertainty across Additional Models.** Solid lines represent Low Entropy (Easy) subsets, and dashed lines represent High Entropy (Hard) subsets.

### F.4 CONSISTENCY OF MONOTONICITY: THE SELECTION OF DIFFERENT UNCERTAINTY INDICES

To verify whether Shannon Entropy is the unique prerequisite for our findings, or if the phenomenon reflects a deeper cognitive state independent of specific mathematical formulations, we extended our evaluation to two alternative uncertainty metrics:

1. **Prediction Margin (Gap):** Defined as $1 - (p_{\text{top1}} - p_{\text{top2}})$. This metric captures the ambiguity between the top two candidates, focusing on conflict intensity.

2. **Negative Log-Probability (NLL):** Defined as $-\log p(y|x)$. This metric directly reflects the model's surprisal or raw confidence in its chosen output.

**A Unified View.** Despite differences in calculation, Entropy, Margin, and NLL serve as unified proxies for the model's **reasoning hesitation**. High values in any of these metrics (or low margin) indicate that the model is struggling to distinguish between modalities.

**Results.** As illustrated in Figure 16, we plot the modality-following curves using these alternative metrics across our **Controlled Dataset**, $MC^2$ **Color Recognition**, and $Pascal$ **VOC Object Recognition**. Crucially, the **strict monotonic decline** is robustly preserved across all datasets and metrics. The curves exhibit the same characteristic shape: as relative uncertainty (regardless of how it is measured) increases, the model's adherence to a modality drops. This consistency confirms that our "Fundamental Law" is driven by the underlying decision dynamics, avoiding the fragility often associated with single-metric analyses.

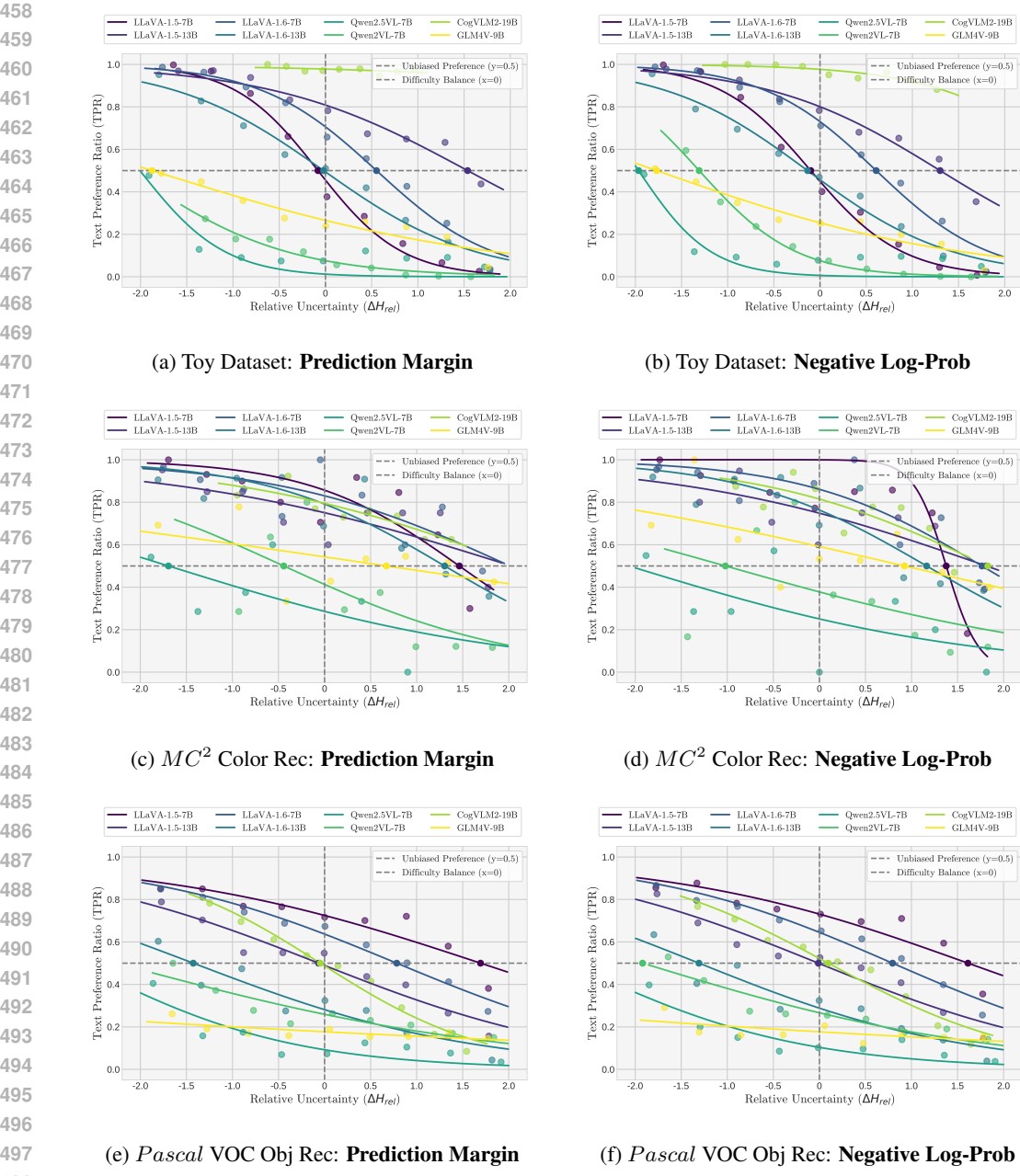

(a) Toy Dataset: **Prediction Margin**

(b) Toy Dataset: **Negative Log-Prob**

(c) $MC^2$ Color Rec: **Prediction Margin**

(d) $MC^2$ Color Rec: **Negative Log-Prob**

(e) $Pascal$ VOC Obj Rec: **Prediction Margin**

(f) $Pascal$ VOC Obj Rec: **Negative Log-Prob**

Figure 16: **Generalization across Uncertainty Metrics.** We replicate the analysis using Prediction Margin (Left Column) and Negative Log-Probability (Right Column) across different datasets. The preservation of the monotonic curve demonstrates that our findings are metric-agnostic and capture the fundamental uncertainty of the models.

## F.5 UNIVERSALITY OF LAYER-WISE OSCILLATION MECHANISM

To validate the generality of our findings regarding the internal conflict resolution mechanism, we extended the layer-wise analysis to all evaluated models across three additional datasets: $MC^2$ **Color Recognition**, $MC^2$ **Object Recognition**, and $Pascal$ **VOC Object Recognition**.

As visualized in Figure 17, Figure 18, and Figure 19, we consistently observe that:

- **High Oscillation in Ambiguity:** Across all tasks and models, the **Ambiguous** subsets (grey bars/lines) exhibit significantly higher oscillation magnitudes compared to the clear Regions.

- **Localized Conflict Resolution:** The layer-wise dynamics figure(right columns) confirm that this "hesitation" is primarily localized to the middle and late layers, where the model integrates conflicting modal information before reaching a final decision.

These extensive experiments confirm that the "oscillation" mechanism is a universal property of MLLMs when resolving multimodal ambiguity.

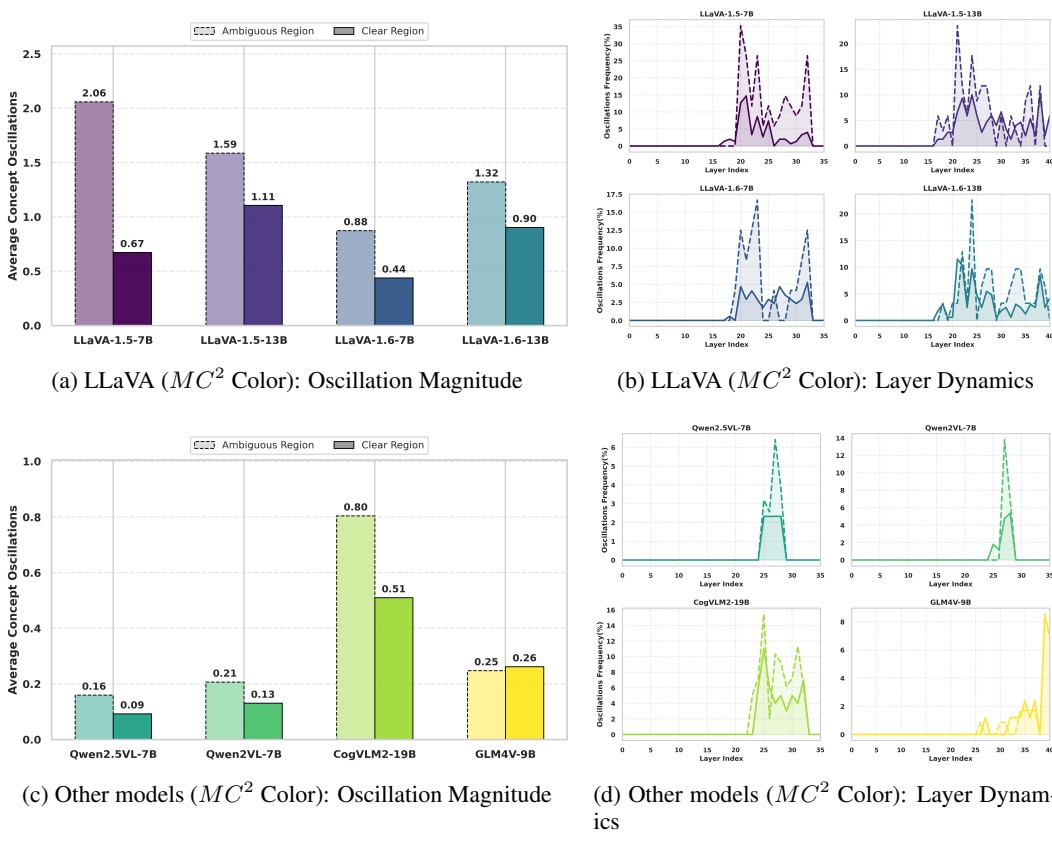

(a) LLaVA ($MC^2$ Color): Oscillation Magnitude

(b) LLaVA ($MC^2$ Color): Layer Dynamics

(c) Other models ($MC^2$ Color): Oscillation Magnitude

(d) Other models ($MC^2$ Color): Layer Dynamics

Figure 17: **Internal Oscillation Analysis on $MC^2$ Color Recognition.** The left column shows the statistical comparison of oscillation magnitude across different uncertainty regions. The right column visualizes the layer-wise evolution of logit differences. Note that the **Ambiguous** region consistently triggers the highest internal conflict.

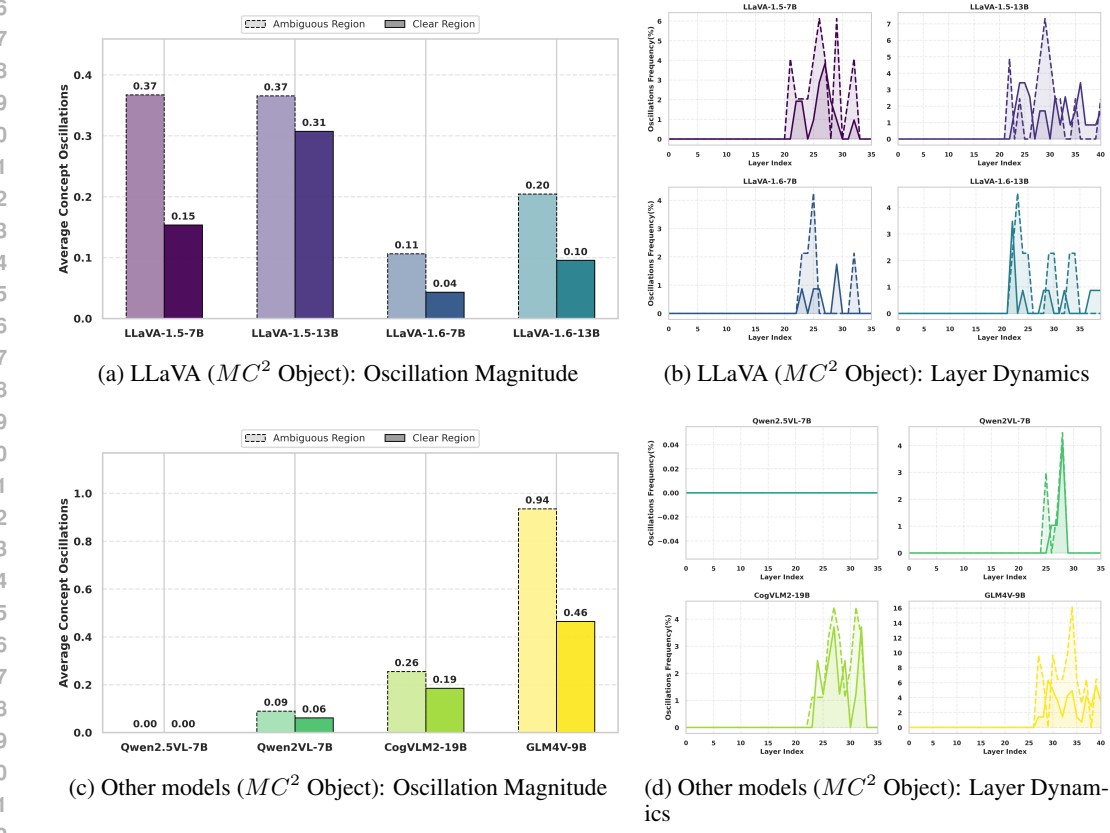

(a) LLaVA ($MC^2$ Object): Oscillation Magnitude

(b) LLaVA ($MC^2$ Object): Layer Dynamics

(c) Other models ($MC^2$ Object): Oscillation Magnitude

(d) Other models ($MC^2$ Object): Layer Dynamics

Figure 18: **Internal Oscillation Analysis on $MC^2$ Object Recognition.** Consistent with color tasks, object recognition tasks also exhibit significant internal oscillation in ambiguous regions, confirming the task-agnostic nature of this mechanism.

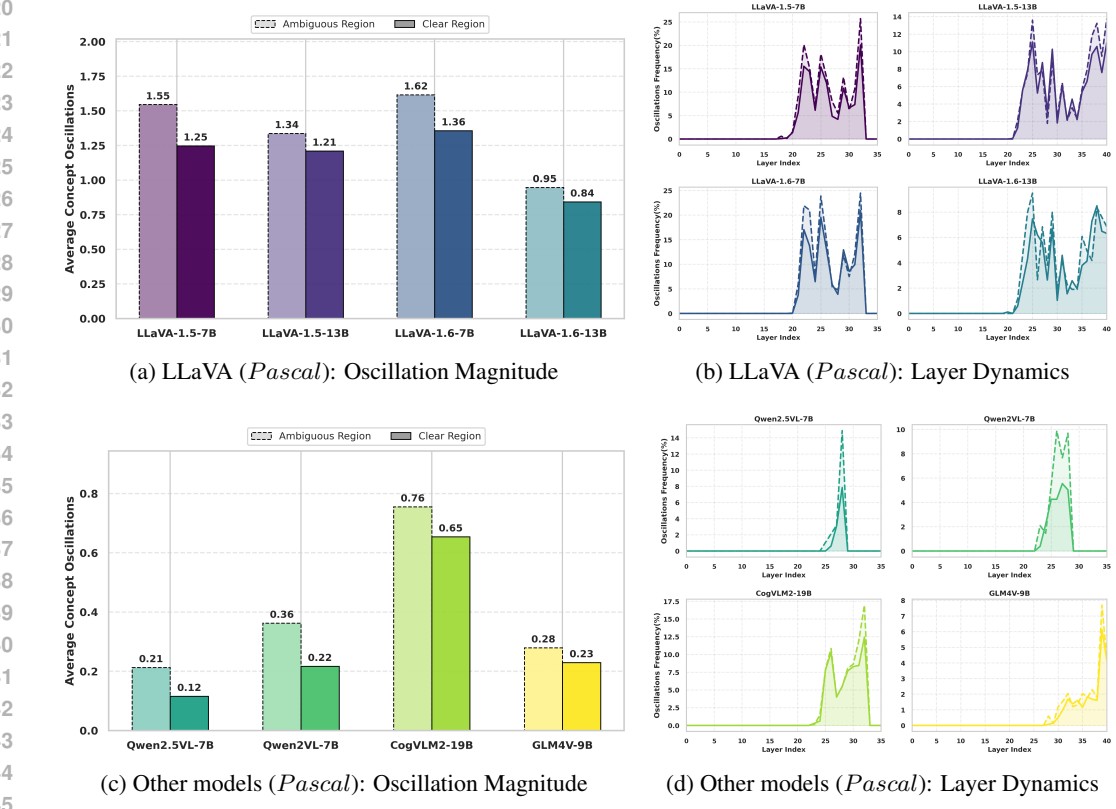

(a) LLaVA ($Pascal$): Oscillation Magnitude

(b) LLaVA ($Pascal$): Layer Dynamics

(c) Other models ($Pascal$): Oscillation Magnitude

(d) Other models ($Pascal$): Layer Dynamics

Figure 19: **Internal Oscillation Analysis on $Pascal$ VOC Object Recognition.** Even in complex real-world datasets like Pascal VOC, the oscillation mechanism remains a robust indicator of multi-modal conflict resolution.

