# OpenReview forum: "When Modalities Conflict: How Unimodal Reasoning Uncertainty Governs Preference Dynamics in MLLMs"
_ICLR.cc/2026/Conference — Submitted to ICLR 2026_

### Official Review · Reviewer_VBB8 · 2025-10-26

**Soundness:** 2
**Presentation:** 3
**Contribution:** 3
**Rating:** 6
**Confidence:** 4

**Summary:**

This paper measures whether vision and language models (VLMs) rather follow the image information or the text information when visual and text inputs contradict each other. The authors propose a framework that decomposes modality following into two factors: relative reasoning uncertainty (the confidence gap between unimodal predictions) and inherent modality preference (a model’s bias when uncertainties are equal). The paper proposes a toy (controllable) dataset and entropy-based uncertainty analysis, they uncover a universal law that a model’s likelihood to follow a modality decreases monotonically with its relative uncertainty, and they link this behavior to internal “oscillations” observed across model layers near ambiguity points.

**Strengths:**

* **Clear and original framing**: The paper presents a compelling ide, namely measuring modality bias by observing which modality an MLLM follows when presented with conflicting information in the two modalities.
* **Strong model coverage**: The authors evaluate a comprehensive set of models spanning diverse LLM backbones (LLaVA and Qwen-based variants), lending credibility and generality to the observed patterns.
* **Mechanistic interpretability insight**: The analysis of internal oscillation patterns under uncertainty is very interesting and valuable mechanistic perspective, loved this.
* **Well-written and accessible**: Despite a few phrasing hiccups in the abstract, the paper is overall clearly structured.
** High relevance and timeliness:** Understanding how MLLMs resolve conflicting multimodal information is highly pertinent to current challenges in multimodal reasoning, perception, and interactive AI systems.

**Weaknesses:**

* **Toy-level dataset and limited task diversity**: The evaluation relies entirely on a synthetic color-recognition setup with simple geometric shapes and textual descriptions. While this design enables tight control over difficulty tiers, it offers little ecological validity. Modality-following behavior may depend on data type, domain, or task semantics, so it remains unclear whether the proposed framework and its findings generalize to real-world multimodal reasoning tasks such as visual QA, caption grounding, or instruction following. Modality following might be data-type and domain-dependent and this work does not show how far this method and its findings generalise.
* **Missing comparison to existing interpretability frameworks for multimodal imbalance:** Although the paper cites prior work such as Parcalabescu & Frank (2022; 2024), it does not meaningfully connect its findings to earlier analyses of modality contribution and cross-modal influence. Important related frameworks such as the Perceptual Score (Gat et al., 2021) and Frank et al. (2021), “Vision-and-language or vision-for-language?” -- all of which reached contradicting conclusions -- are not discussed and not compared to, leaving the paper insufficiently situated within the broader interpretability and multimodal reasoning literature.
* **Entropy-based uncertainty analysis may not be robust**: The framework hinges on token-level output entropy as a proxy for reasoning uncertainty. However, entropy primarily captures surface-level distributional sharpness rather than true epistemic or aleatoric uncertainty, and it can be influenced by decoding strategies or output vocabulary size or task ambiguity. Without calibration checks or comparison to alternative uncertainty estimates, the reliability of the conclusions drawn from entropy remains uncertain.

**Questions:**

The sentence starting in L015 is confusingly framed. It reads: "Prior work measured this behavior only with coarse dataset-level statistics, overlooking the influence of models’ confidence in unimodal reasoning." It is unclear what "dataset-level" relates to with overlooking a completely new object of measurement, namely confidence in unimodal reasoning.

L018-L019 parentheses are broken because there is a whitespace after them → remove the whitespace

L021: "using entropy..." → entropy of what?

---

> ### Author Response · Authors · 2025-11-23
> **Reply to Reviewer VBB8 on W1 & W3**
>
> ## Reply to Reviewer VBB8 on W1 & W3
>
> **[W1] Generalization to Real-World Multimodal Tasks (Addressing "Toy-level dataset")**
>
> > The evaluation relies entirely on a synthetic color-recognition setup with simple geometric shapes and textual descriptions. While this design enables tight control over difficulty tiers, it offers little ecological validity. Modality-following behavior may depend on data type, domain, or task semantics, so it remains unclear whether the proposed framework and its findings generalize to real-world multimodal reasoning tasks such as visual QA, caption grounding, or instruction following. Modality following might be data-type and domain-dependent and this work does not show how far this method and its findings generalise.
>
> Thank you for the reviewer for these critical observations. We have **significantly expanded our experimental scope** in the revised manuscript to demonstrate that our framework generalizes to real-world scenarios and is robust across different uncertainty quantification methods in **Section 4**:
> We evaluated models on the **$MC^2$ benchmark** (involving real-world photography for Object and Color Recognition) [1] and a conflict-modified **Pascal VOC dataset** (focused on semantic Object Recognition) [2]. These tasks involve complex visual scenes and high-level semantic conflicts (e.g., distinguishing between a "cow" and an "eagle" in a natural background).
>
> **Results:**
>
> - **Universal Behavioral Law:** As shown in **Figure 4(a-c)** and **Figure 14**, we found that the "Relative Uncertainty Law" is domain-agnostic. The probability of following a modality monotonically decreases as its relative uncertainty increases, regardless of whether the task involves simple geometric attributes or complex semantic objects.
> - **Universal Internal Mechanism:** Furthermore, in **Section 5** and **Appendix F.5**, we verified that the internal "oscillation" mechanism is also present and strictly localized to middle-to-late layers in these real-world tasks (**Figures 17, 18, 19**).
>
> ---
>
> **[W3] Robustness of Uncertainty Analysis (Addressing "Entropy-based analysis")**
>
> > **Entropy-based uncertainty analysis may not be robust**: The framework hinges on token-level output entropy as a proxy for reasoning uncertainty. However, entropy primarily captures surface-level distributional sharpness rather than true epistemic or aleatoric uncertainty, and it can be influenced by decoding strategies or output vocabulary size or task ambiguity. Without calibration checks or comparison to alternative uncertainty estimates, the reliability of the conclusions drawn from entropy remains uncertain.
>
> To address the concern that token-level entropy might be a fragile proxy, we conducted extensive robustness checks in **Section 4** and **Appendix F.4**:
>
> * **Validation with Alternative Metrics:** We replicated our core analysis using two alternative metrics that capture different aspects of uncertainty:
>   1.  **Prediction Margin (Gap):** Captures the conflict intensity between the top candidates ($1 - (p_{top1} - p_{top2})$).
>   2.  **Negative Log-Probability (NLL):** Reflects the model's raw surprisal ($- \log p(y|x)$).
>
> **Results:**
>
> - As illustrated in **Figure 16**, the strict monotonic decline in modality following is robustly preserved across all datasets and all three metrics.  This confirms that our findings are not artifacts of the entropy formula but reflect the model's fundamental underlying uncertainty.
>
> These new experiments confirm that our framework is not limited to toy datasets or specific metrics but represents a generalizable principle of multimodal conflict resolution.
>
> > [1] Evaluating and Steering Modality Preferences in Multimodal Large Language Model
> > [2] How Do Vision-Language Models Process Conflicting Information Across Modalities?

---

> > ### Comment · Reviewer_VBB8 · 2025-11-23
> > **Reply to W1 & W2 & W3**
> >
> > Thanks for everything you added to the paper, I increase my score.

---

> ### Author Response · Authors · 2025-11-23
> **Reply to Reviewer VBB8 on W2**
>
> ## Reply to Reviewer VBB8 on W2
> **[W2]**
> > Missing comparison to existing interpretability frameworks for multimodal imbalance: Although the paper cites prior work such as Parcalabescu & Frank (2022; 2024), it does not meaningfully connect its findings to earlier analyses of modality contribution and cross-modal influence. Important related frameworks such as the Perceptual Score (Gat et al., 2021) and Frank et al. (2021), “Vision-and-language or vision-for-language?” -- all of which reached contradicting conclusions -- are not discussed and not compared to, leaving the paper insufficiently situated within the broader interpretability and multimodal reasoning literature.
> >
> In the revised **Related Work** section, we have added a dedicated discussion on “the linguistic priors learned during pre-training are conceptually central to modality preference”.
>
> We explicitly discuss works like [1] and [2] regarding modality imbalance, and [3] regarding the origins of "visual priors" from text pre-training. These static linguistic priors fundamentally shape what we term the model's "inherent preference."
> However, we advance this line of inquiry by proposing that modality following is not a fixed attribute dominated solely by these priors. Instead, we frame it as a **fluid behavior** governed by **Relative Reasoning Uncertainty**. In our framework, the "inherent preference" (shaped by pre-training) serves as a baseline balance point, which is dynamically modulated by the case-specific confidence gap between unimodal reasoning paths.
>
> We believe this discussion effectively bridges the gap between the static capabilities identified in prior literature and the dynamic decision-making dynamics we observe during conflict resolution.
>
> > [1] Perceptual score: What data modalities does your model perceive?
> >
> > [2] Vision-and-language or vision-for-language? on cross-modal influence in multimodal transformers.
> >
> > [3] Learning to see before seeing: Demystifying llm visual priors from language pretraining

---

> ### Author Response · Authors · 2025-11-23
> **Reply to Reviewer VBB8 on Q1**
>
> ## Reply to Reviewer VBB8 on Q1
> **[Q1]**
> > The sentence starting in L015 is confusingly framed. It reads: "Prior work measured this behavior only with coarse dataset-level statistics, overlooking the influence of models’ confidence in unimodal reasoning." It is unclear what "dataset-level" relates to overlooking a completely new object of measurement, namely confidence in unimodal reasoning.
>
> The logical connection is that **dataset-level aggregation inherently masks instance-level variance**, leading to a critical confounding of "capability" with "preference."
>
> Prior work relied on global metrics (e.g., Text-Following Ratios) to characterize behavior. However, reasoning confidence is a dynamic, **instance-level variable** that fluctuates significantly from case to case. By collapsing these fluctuations into a single average, dataset-level statistics smooth out the signal of confidence, making it impossible to observe its governing role.
>
> This aggregation leads to misinterpretation. For example, as shown in **Section 4**, **Qwen2-VL** appears to have a strong "vision preference" based on dataset-level statistics. However, our instance-level analysis reveals this is an artifact of its strong visual **capability** (low visual uncertainty) on this specific dataset. When we account for confidence, we discover that Qwen2-VL actually possesses an inherent preference different from what the aggregate stats suggest. Dataset-level measurement overlooked this distinction entirely.

---

> > ### Comment · Reviewer_VBB8 · 2025-11-23
> > **Reply to discussion about Q1**
> >
> > Thanks a lot for the detailed reply. I’m still not sure how a reader of the abstract is supposed to clearly see the distinction between "coarse dataset-level statistics" and "influence of models' confidence in unimodal reasoning". I do understand what you did, and my point in Q1 was just that this isn’t formulated clearly for an abstract.
> >
> > For instance, someone could easily read "coarse dataset-level statistics" as referring to statistics over confidence of individual samples (or something else entirely). And when you say "However, our instance-level analysis reveals this is an artifact of its strong visual capability (low visual uncertainty) on this specific dataset", you are averaging over instances to infer strong visual capability, no?
> >
> > I'm just saying that verbally, the distinction is not clear and the writing and explanation can be improved to be more didactic and clear.

---

> > > ### Author Response · Authors · 2025-11-24
> > > **Reply to the Q1 about the writing and explanation**
> > >
> > > Thank you very much for your response and for the helpful clarification. We appreciate your detailed suggestion regarding the distinction between coarse dataset-level statistics and instance-level confidence effects in the abstract and introduction. We agree that the current wording may cause ambiguity for readers, and **we will refine the explanation in the next revision** to make the distinction clearer.

---

### Official Review · Reviewer_yL48 · 2025-10-28

**Soundness:** 3
**Presentation:** 3
**Contribution:** 2
**Rating:** 4
**Confidence:** 3

**Summary:**

This paper investigates "modality following" in MLLMs, a phenomenon describing how models resolve conflicts between contradictory visual and textual information. The authors argue that prior work, which relies on coarse, macro-level statistics, overlooks the critical role of a model's case-specific uncertainty in its unimodal reasoning. To address this, the paper introduces an innovative analytical framework that decomposes modality following into two core factors: (1) relative reasoning uncertainty, which is the case-specific confidence gap between predictions based on vision versus text alone, and (2) inherent modality preference, a model's stable, intrinsic bias toward one modality when uncertainties are balanced. To validate this framework, the authors construct a controllable dataset that systematically varies the reasoning difficulty of both visual and textual inputs. Using output entropy as a quantitative measure of uncertainty, the paper discovers a "unified monotonic law": the probability of a model following a modality decreases monotonically as its relative uncertainty increases. This law allows the authors to define the "balance point"—the relative uncertainty value at which a model is equally likely to follow either modality—as a principled, quantitative metric for its inherent preference.

**Strengths:**

1. The paper introduces a novel and principled analytical framework that deconstructs the complex phenomenon of modality following into two more fundamental components: relative reasoning uncertainty and inherent modality preference. This approach moves beyond the prior work to offer a more powerful explanatory model, representing a conceptual contribution to understanding multimodal conflict resolution.
2. The work is supported by a relatively rigorous experimental design, centered on a custom-built dataset where visual and textual reasoning difficulty can be independently and systematically controlled. This allows for the precise isolation and study of the core variables (uncertainties) and their effect on model behavior, setting a standard for mechanistic and interpretability research in large models.
3. The empirical discovery of a "unified monotonic law" is a simple yet useful finding that organizes seemingly chaotic model choices into a single, predictable pattern. The "balance point" concept that emerges from this law provides a principled method for quantifying and disentangling a model's intrinsic biases from confounding factors like its unimodal capabilities or dataset-specific artifacts.

**Weaknesses:**

1. The core findings are predominantly derived from a synthetic "toy dataset" focused on color and attribute recognition of simple geometric shapes. While this controlled setting is ideal for validating fundamental principles, it raises questions about the generalizability of the conclusions to more complex, subtle, and semantic conflicts found in real-world scenarios. It is not yet clear if the framework, especially the stability of a model's "balance point," holds across a wider variety of realistic tasks.
2. The entire analysis relies heavily on using output token entropy as the sole proxy for model uncertainty. Although entropy is shown to correlate with designed difficulty, the paper does not sufficiently justify this choice over other uncertainty quantification methods or discuss its potential limitations. The robustness of the findings is therefore contingent on the validity of this single metric.
3. The experiments are conducted on a limited set of six MLLMs from the two families. While these are relevant models, they represent a narrow slice of the architectural landscape. The claim of discovering a "universal law" would be significantly strengthened by validating the framework on a more diverse set of models with different architectures and training paradigms.
4. The paper provides a strong diagnostic framework but offers limited discussion on its practical implications for improving models. For example, it is unclear how this understanding could be used to steer a model's modality preference during alignment or to develop methods that reduce internal oscillations and make models more decisive.

**Questions:**

1. How do you envision your framework applying to more open-ended, generative tasks where a model must produce long-form text? In such scenarios, the entropy of a single token may be insufficient to capture the model's overall uncertainty. How would the concept of "relative uncertainty" need to be adapted or extended for these more complex cases?
2. Regarding the "internal oscillations," have you analyzed their distribution across the model's layers? For instance, do oscillations occur more frequently in early, middle, or late layers of the network? Does the magnitude of the oscillations correlate with the overall model uncertainty?

---

> ### Author Response · Authors · 2025-11-23
> **Reply to Reviewer yL48 on W1 & W2 & W3**
>
> ## Reply to Reviewer yL48 on W1 & W2 & W3 & W4 & Q2
>
> Thank you for these insightful comments. In response, we have significantly expanded our experimental scope to demonstrate the robustness, universality, and practical utility of our framework. Below, we address each point using the new data and analyses included in the revised manuscript.
>
> **[W1] Generalizability to Real-World Scenarios**
>
> > The core findings are predominantly derived from a synthetic "toy dataset" focused on color and attribute recognition of simple geometric shapes. While this controlled setting is ideal for validating fundamental principles, it raises questions about the generalizability of the conclusions to more complex, subtle, and semantic conflicts found in real-world scenarios. It is not yet clear if the framework, especially the stability of a model's "balance point," holds across a wider variety of realistic tasks.
>
> We explicitly addressed this by extending our evaluation to two challenging real-world benchmarks in **Section 4**: the **$MC^2$ benchmark** (involving real-world object and color recognition) [1]and a conflict-modified **Pascal VOC dataset** [2].
> **Results:**
>
> - As shown in **Figure 4(a-c)** and **Appendix F.2** (Figure 14), the fundamental law—that modality following monotonically decreases with relative uncertainty—holds robustly across these complex semantic tasks.
> - **Balance Point Stability:** We confirmed that while the *position* of the balance point shifts depending on the task (reflecting the model's inherent preference for that specific domain), the *stability* of the preference dynamics remains consistent across diverse realistic tasks.
>
> ---
>
> **[W2]Robustness of Entropy as a Metric**
>
> > The entire analysis relies heavily on using output token entropy as the sole proxy for model uncertainty. Although entropy is shown to correlate with designed difficulty, the paper does not sufficiently justify this choice over other uncertainty quantification methods or discuss its potential limitations. The robustness of the findings is therefore contingent on the validity of this single metric.
>
> We validated the robustness of our findings against alternative uncertainty metrics in **Section 4** and **Appendix F.4**.
> We replicated our analysis using **Prediction Margin** (capturing conflict intensity) and **Negative Log-Probability** (reflecting raw surprisal).
>
> **Results:**
>
> - As illustrated in **Figure 16**, the monotonic relationship between uncertainty and modality following is preserved regardless of the metric used. This confirms that our framework captures the underlying cognitive state of uncertainty rather than being an artifact of the entropy formula.
>
> Furthermore, we have added a new Section 7, DISCUSSION AND FUTURE WORK, to discuss our further views on this indicator.
>
> ---
>
> **[W3] Diversity of Model Architectures**
>
> > The experiments are conducted on a limited set of six MLLMs from the two families. While these are relevant models, they represent a narrow slice of the architectural landscape. The claim of discovering a "universal law" would be significantly strengthened by validating the framework on a more diverse set of models with different architectures and training paradigms.
>
> We expanded our model suite to include **CogVLM2** and **GLM-4V**, bringing the total to 8 models across diverse architectural paradigms (e.g., Deep Fusion vs. MLP Projection).
> **Table 5** in **Appendix D** details the structural differences among these families.
>
> **Results:**
>
> - The results in **Figure 3** and **Figure 4** confirm that these distinct architectures all adhere to the same "Relative Uncertainty Law," significantly strengthening our claim of universality.
>
>
>
> >[1] Evaluating and Steering Modality Preferences in Multimodal Large Language Model
> >[2]  How Do Vision-Language Models Process Conflicting Information Across Modalities?

---

> ### Author Response · Authors · 2025-11-23
> **Reply to Reviewer yL48 on W4 & Q2**
>
> **[W4] Practical Implications for Steering and Alignment**
>
> > The paper provides a strong diagnostic framework but offers limited discussion on its practical implications for improving models. For example, it is unclear how this understanding could be used to steer a model's modality preference during alignment or to develop methods that reduce internal oscillations and make models more decisive.
>
> We added a completely new section, **Section 6: Application: Guiding Data Selection for Preference Steering**, to address this.
>
> We demonstrated a Supervised Fine-Tuning (SFT) experiment where we used our $\Delta H_{rel}$ metric to select training data.
>
> **Results:**
>
> - We found that training on "easy" samples fails to generalize, whereas targeting ambiguous and hard-negative regions identified by our metric is essential for robust preference alignment.
> - **Reducing Oscillation:** As shown in **Figure 7b**, models trained on these hard/ambiguous samples showed a significant **reduction in internal oscillations** (negative bars), effectively making the models more decisive.
>
> ---
>
> **[Q2] Analysis of Internal Oscillations**
>
> > *Regarding the "internal oscillations," have you analyzed their distribution across the model's layers? For instance, do oscillations occur more frequently in early, middle, or late layers of the network? Does the magnitude of the oscillations correlate with the overall model uncertainty?*
>
> We provided a detailed layer-wise analysis in **Section 5** and **Appendix F.5**.
>
> **Results:**
>
> - **Layer Distribution:** As shown in **Figure 5b** and **Figure 17 bd,Figure 18 bd,Figure 19 bd**, We found that oscillations are not uniform but are strictly **localized to the middle-to-late layers** (typically after layer 15). This suggests conflict resolution is a high-level cognitive process.
> - **Correlation with Uncertainty:** As shown in **Figure 5a** and **Figure 17 ac,Figure 18 ac,Figure 19 ac** We explicitly analyzed the magnitude of oscillations across different uncertainty regions. the oscillation frequency is significantly higher in the **Ambiguous Region** (where relative uncertainty is high)

---

> ### Author Response · Authors · 2025-11-23
> **Reply to Reviewer yL48 on Q1**
>
> ## Reply to Reviewer yL48 on Q1
>
> > [Q1] How do you envision your framework applying to more open-ended, generative tasks where a model must produce long-form text? In such scenarios, the entropy of a single token may be insufficient to capture the model's overall uncertainty. How would the concept of "relative uncertainty" need to be adapted or extended for these more complex cases?
>
> We posit that our focus on short-answer tasks is a strategic choice to isolate the fundamental law of conflict resolution, which serves as the atomic unit of decision-making even in complex scenarios. In open-ended generation (e.g., Chain-of-Thought), conflicting information does not disappear but acts as a "pivotal node" within a longer reasoning chain; thus, the relative uncertainty at these nodes continues to govern the branching of reasoning paths. Consequently, applying our framework to these contexts does not require changing the underlying logic, but rather adapting the measurement granularity. As discussed in **New Section 7**, future work should move beyond token-level entropy to employ advanced metrics like Semantic Entropy or aggregated sequence probabilities, which allow for the quantification of relative uncertainty over entire concepts or reasoning steps effectively

---

> ### Author Response · Authors · 2025-11-24
> **Updated PDF & Rebuttal Submitted**
>
> Our rebuttal and updated PDF are now available. Please let us know whether the response resolves your concerns, and if so, whether it affects your score—or if there are remaining questions to address.

---

> > ### Comment · Reviewer_yL48 · 2025-11-26
> >
> > I appreciate the authors’ efforts and the additional clarifications provided in the rebuttal. However, I would like to maintain my original evaluation.

---

> ### Author Response · Authors · 2025-11-27
> **Reply to Reviewer yL48**
>
> We genuinely believe our revised PDF effectively addresses your concerns. If you have no further questions, we respectfully ask if you would consider raising your score to reflect these improvements.

---

### Official Review · Reviewer_x1XN · 2025-10-30

**Soundness:** 3
**Presentation:** 3
**Contribution:** 3
**Rating:** 6
**Confidence:** 5

**Summary:**

This paper investigates how multimodal models decide whether to follow textual or visual information when the two modalities provide conflicting cues. The authors propose a principled framework that decomposes this “modality-following” behavior into two underlying components: relative unimodal reasoning uncertainty and the model’s inherent modality preference.

To test this framework, they design a controllable toy dataset that allows systematic manipulation of reasoning difficulty in each modality, thereby inducing varying uncertainty levels. By quantifying uncertainty via the output entropy of answer tokens, the study establishes a clear empirical relationship: the probability that a model follows a modality monotonically decreases as that modality’s uncertainty increases. Moreover, each model exhibits a subjective balance point, a stable threshold reflecting its internal preference toward vision or text.

At the mechanistic level, the authors uncover layer-wise oscillations in ambiguous conditions, where the model alternates between textual and visual answers before converging. This oscillation provides a plausible explanation for externally observed indecision

**Strengths:**

- The finding that the probability of following a modality decreases monotonically as its relative uncertainty increases is interesting!
- The paper provides a clear and interpretable decomposition of multimodal decision-making, distinguishing between case-specific uncertainty and model-level bias.
- The paper’s analysis is systematic, the hypotheses are well-motivated, and the results are supported by clear empirical trends and visualizations

**Weaknesses:**

- Some prior works on semantic bias, language bias might worth discussing. For instance, [1] argue that linguistic priors learned during pre-training can “hack” or dominate visual inference, which appears conceptually related to the present work’s notion of modality preference.
- Model selection for Figure 4(a) and 4(b).
Different model sets are used in these two subfigures, but it is unclear whether the remaining models exhibit similar behavioral patterns as the three presented ones.
- Model identification in Figure 6(a).
The paper does not specify which model generated the visualization in Figure 6(a). It remains uncertain whether the observed layer-wise oscillation behavior is consistent across different model architectures or specific to one instance.

[1] Han, Junlin, Shengbang Tong, David Fan, Yufan Ren, Koustuv Sinha, Philip Torr, and Filippos Kokkinos. "Learning to See Before Seeing: Demystifying LLM Visual Priors from Language Pre-training." arXiv preprint arXiv:2509.26625 (2025).

**Questions:**

- How the average concept oscillations in figure 5 is computed ? For example, we have 500 samples, and for each sample i, model has x_i layer-wise oscillations, is it computed as \sum_{I}{x_i}/500 or \sum_{I}{x_i}/layer number/500 or other ways?


- Expand Literature Discussion on Semantic and Language Biases.
- Provide details on all models analyzed in Figure 4 and indicate whether unreported models display comparable patterns.
- If some models diverge from the main trend, discussing these exceptions would offer valuable insights into when and why the proposed law holds.
- Identify the specific model used to produce the layer-wise oscillation visualization in Figure 6(a).
- Include (or mention in the appendix) parallel analyses for other models to demonstrate the robustness and generality of this internal oscillation phenomenon.

---

> ### Author Response · Authors · 2025-11-23
> **Reply to Reviewer x1XN on W2 & W3 & Q3 & Q6**
>
> ## Reply to Reviewer x1XN on W2 & W3 & Q3 & Q6
>
> > [W2] Model selection for Figure 4(a) and 4(b). Different model sets are used in these two subfigures, but it is unclear whether the remaining models exhibit similar behavioral patterns as the three presented ones.
> >
> > [W3] Model identification in Figure 6(a). The paper does not specify which model generated the visualization in Figure 6(a). It remains uncertain whether the observed layer-wise oscillation behavior is consistent across different model architectures or specific to one instance.
> >
> > [Q3] Provide details on all models analyzed in Figure 4 and indicate whether unreported models display comparable patterns.
> >
> > [Q6] Include (or mention in the appendix) parallel analyses for other models to demonstrate the robustness and generality of this internal oscillation phenomenon.
>
> We thank the reviewer for the constructive suggestion to clarify our model selection and demonstrate the generality of our findings. In the revised manuscript, we have organized our analysis to present representative results in the main text while providing extensive **parallel experiments** in **Appendix F** to ensure full coverage of all evaluated models (LLaVA-1.5/1.6 families, Qwen2/2.5-VL, CogVLM2, and GLM-4V) and datasets.
>
> Below, we detail these parallel analyses, linking the main text findings to their comprehensive validations in the Appendix:
>
> 1. Parallel Experiments on Behavioral Consistency (Extending Figure 4)
>
> - **Figure 4** illustrates the "Relative Uncertainty Law" and "Robustness to Absolute Difficulty" primarily using the Controlled Dataset and representative 7B models (LLaVA-1.5-7B, LLaVA-1.6-7B, Qwen2VL-7B).
>
>   - **Full Model Extension (Appendix F.1, Figure 13):** We extend the macro-level performance and uncertainty distribution analysis to **all 8 evaluated models** on the real-world $MC^2$ and Pascal VOC datasets. This confirms that models not fully detailed in the main text (e.g., CogVLM2, GLM-4V) exhibit consistent behavioral patterns
>
>   - **Absolute Difficulty Robustness (Appendix F.3, Figure 15):** We replicate the "High vs. Low Entropy" analysis for larger and newer models, specifically **LLaVA-1.5-13B, LLaVA-1.6-13B, and Qwen2.5-VL-7B**. This confirms that the interaction between absolute and relative uncertainty holds across model scales.
>
>   - **Metric Independence (Appendix F.4, Figure 16):** We replicate the behavioral curves using alternative metrics (**Prediction Margin** and **Negative Log-Probability**) for all models, ensuring the findings are not specific to Entropy.
>
> ----
>
> 2. Parallel Experiments on Internal Mechanism (Extending Figure 6)
>
> - **Figure 6(a)** visualizes the layer-wise oscillation mechanism.  We explicitly clarify that this figure displays the **LLaVA-1.5 (7B/13B)** and **LLaVA-1.6 (7B/13B)** families, serving as our primary case study for the internal dynamics of indecision.
>   -  **Architecture Generalization (Appendix F.5, Figures 17, 18, 19):** To address the concern about whether this behavior is "specific to one instance," we conducted parallel layer-wise probing for **all non-LLaVA models** (Qwen2-VL, Qwen2.5-VL, CogVLM2, and GLM-4V)
>
>      - **Task Generalization:** These parallel experiments were performed across three distinct datasets ($MC^2$ Color, $MC^2$ Object, and Pascal VOC) in Appendix F.5, Figures 17, 18, 19. The results consistently show that high-frequency oscillation in ambiguous regions is a universal mechanism across diverse architectures
>
> These parallel experiments confirm that the behavioral laws and internal mechanisms reported in the main text are robust, universal, and not limited to specific model selections.

---

> ### Author Response · Authors · 2025-11-23
> **Reply to Reviewer x1XN on W1 & Q2 & Q1 & Q5**
>
> ## Reply to Reviewer x1XN on W1 & Q2 & Q1 & Q5
> **[W1 & Q2]**
> > [W1] Some prior works on semantic bias, language bias might worth discussing. For instance, [1] argue that linguistic priors learned during pre-training can “hack” or dominate visual inference, which appears conceptually related to the present work’s notion of modality preference.
> >
> > [Q2]Expand Literature Discussion on Semantic and Language Biases.
>
> In the revised **Related Work** section, we have added a dedicated discussion on “the linguistic priors learned during pre-training are conceptually central to modality preference”.
>
> We explicitly discuss works like [1] and [2] regarding modality imbalance, and [3] regarding the origins of "visual priors" from text pre-training. These static linguistic priors fundamentally shape what we term the model's "inherent preference."
> However, we advance this line of inquiry by proposing that modality following is not a fixed attribute dominated solely by these priors. Instead, we frame it as a **fluid behavior** governed by **Relative Reasoning Uncertainty**. In our framework, the "inherent preference" (shaped by pre-training) serves as a baseline balance point, which is dynamically modulated by the case-specific confidence gap between unimodal reasoning paths.
>
> We believe this discussion effectively bridges the gap between the static capabilities identified in prior literature and the dynamic decision-making dynamics we observe during conflict resolution.
>
> ------
>
> **[Q1]**
>
> > [Q1] How the average concept oscillations in figure 5 is computed ? For example, we have 500 samples, and for each sample i, model has x_i layer-wise oscillations, is it computed as \sum_{I}{x_i}/500 or \sum_{I}{x_i}/layer number/500 or other ways?
>
> **1. Calculation of Average Concept Oscillations**
>
> The "Average Concept Oscillations" in **Figure 5(a)** is computed as the **average number of oscillation events per sample**. Using the reviewer's notation, if we have $N=500$ samples, and for each sample $i$, the model exhibits $x_i$ layer-wise oscillations (total switches between vision/text answers during the forward pass), the value is computed as:
> $$\text{Average Oscillations} = \frac{\sum_{i=1}^{N} x_i}{N}$$
> This metric represents the average "instability" or number of decision flips a single input undergoes as it propagates through the network. It is **not** normalized by the number of layers, as our goal is to quantify the total cognitive struggle per inference instance.
>
> **2. Expanded Experimental Validation**
> To ensure the robustness of this metric and the observed phenomenon, we have significantly expanded our analysis in the revised manuscript. As detailed in **Figures 5, 17, 18, and 19**, we consistently observe that the average oscillation count is significantly higher in the **Ambiguous Region** compared to the Clear Region across all models and tasks. This confirms that the oscillation magnitude is a universal and reliable internal correlate of decision uncertainty.
>
> ------
>
> **[Q5]**
>
> > [Q5] Identify the specific model used to produce the layer-wise oscillation visualization in Figure 6(a).
>
> We have already identified the specific model used in every figure and result. The source of case study is selected from the results of LLaVA-1.5-7B.
>
>
>
> > [1] Perceptual score: What data modalities does your model perceive?
> >
> > [2] Vision-and-language or vision-for-language? on cross-modal influence in multimodal transformers.
> >
> > [3] Learning to see before seeing: Demystifying llm visual priors from language pretraining

---

### Official Review · Reviewer_VpJP · 2025-11-08

**Soundness:** 3
**Presentation:** 2
**Contribution:** 2
**Rating:** 4
**Confidence:** 2

**Summary:**

This paper studies how MLLMs handle conflicting information between text and images. It shows that modality-following behavior arises from two key factors: relative reasoning uncertainty (confidence gap between unimodal predictions) and inherent modality preference (a model’s stable bias). Using a controllable dataset and entropy-based uncertainty measures, the authors find a monotonic law—the likelihood of following a modality decreases as its relative uncertainty grows—and identify layer-wise oscillations near the balance point, explaining indecision. The work provides a unified and interpretable framework for understanding modality preference in MLLMs.

**Strengths:**

1. Novel conceptual framework – Clearly decomposes modality-following behavior into relative uncertainty and inherent preference, offering a principled explanation beyond prior dataset-level analyses.
2. Mechanistic insight – The layer-wise oscillation analysis provides an interpretable link between internal model dynamics and external behavioral indecision.
3. Practical interpretability – The notion of a “balance point” gives a simple quantitative metric for comparing inherent modality preferences across models.

**Weaknesses:**

1. Limited scenario diversity – Both experimental settings focus on geometric shape perception tasks with relatively simple and synthetic visual scenes. This narrow scope limits the generalizability of the findings to more complex, real-world multimodal reasoning scenarios.
2. Lack of downstream validation —— The paper does not demonstrate whether understanding or controlling modality preference improves practical tasks (e.g., VQA or captioning).
3. Interpretability gap —— While oscillation is shown as an internal correlate of indecision, causal evidence connecting it to model training or architecture is still limited.
4. Figure formatting issue – In Figure 4, the font sizes and label alignments are inconsistent, which slightly affects readability and visual coherence.

**Questions:**

1. Your analysis elegantly decomposes modality-following behavior into relative uncertainty and inherent preference. However, how can this framework inform practical improvements in multimodal model design or training?
2. The experiments mainly validate the correlation between H^{(t)} (text entropy) and accuracy under unimodal (T, Q) inputs. However, this only demonstrates that entropy reflects uncertainty in isolated textual reasoning. How does this directly justify its use as the governing variable for modality-following behavior in the conflict setting, where cross-modal interactions and contextual dependencies might differ?

---

> ### Author Response · Authors · 2025-11-23
> **Reply to Reviewer VpJP on W1**
>
> ### [W1] Limited scenario diversity
>
> > Limited scenario diversity – Both experimental settings focus on geometric shape perception tasks with relatively simple and synthetic visual scenes. This narrow scope limits the generalizability of the findings to more complex, real-world multimodal reasoning scenarios.
>
> We thank the reviewer for highlighting the importance of scenario diversity. We agree that relying solely on synthetic geometric tasks limits generalizability. In response, we have **significantly expanded our experimental scope** to include diverse real-world benchmarks and additional model families. Our updated results demonstrate that our core findings, both the behavioral "Relative Uncertainty Law" and the internal "Oscillation Mechanism", hold robustly across complex, real-world scenarios.
>
> 1. **Expansion to Real-World Benchmarks and Diverse Tasks (Section 4)**
>    As shown in **Figure 4 (a-c)**, we integrated challenging real-world datasets adapted for conflict analysis:
>     - **$MC^2$ Benchmark** : We evaluated models on **Object Recognition** and **Color Recognition** tasks from the $MC^2$ dataset released in  [1]. These tasks involve real-world photography and require semantic understanding of complex scenes.
>     - **Pascal VOC (Conflict-Modified)**: Follow [2], we adapted the Pascal VOC dataset to create a new conflict benchmark focused on **Object Recognition**. This introduces conflicts in natural images (e.g., distinguishing trains vs. birds), testing the model's ability to handle high-level semantic contradictions.
>     - **New Model Families:** We expanded our evaluation suite to include **CogVLM2** and **GLM-4V** alongside the LLaVA and Qwen families, ensuring our findings cover diverse architectural strategies.
>
>
> 2. **Universality of the Internal Oscillation Mechanism (Section 5)**
>    Crucially, we demonstrated that our mechanistic interpretability findings,that hesitation stems from internal oscillations,extend to these real-world scenarios:
>     - **Mechanism Verification:** In **Section 5** and **Appendix F.5**, we extended our layer-wise probing to the $MC^2$ and Pascal VOC datasets.
>
> >[1] Evaluating and Steering Modality Preferences in Multimodal Large Language Model
> >[2]  How Do Vision-Language Models Process Conflicting Information Across Modalities?

---

> > ### Author Response · Authors · 2025-11-23
> > **Reply to Reviewer VpJP on W2 & W3 & Q1**
> >
> > **[W2 & W3 & Q1]**
> >
> > > [W2] Lack of downstream validation —— The paper does not demonstrate whether understanding or controlling modality preference improves practical tasks (e.g., VQA or captioning).
> > >
> > > [W3] Interpretability gap —— While oscillation is shown as an internal correlate of indecision, causal evidence connecting it to model training or architecture is still limited.
> > >
> > > [Q1] Your analysis elegantly decomposes modality-following behavior into relative uncertainty and inherent preference. However, how can this framework inform practical improvements in multimodal model design or training?
> >
> > We thank the reviewer for these insightful questions.
> >
> > To slove the questions of the practical applications of our framwork, we add Section 6 **"APPLICATION: GUIDING DATA SELECTION FOR PREFERENCE STEERING"**, we explicitly address these concerns by applying our framework to a practical downstream task: Preference Steering via Supervised Fine-Tuning (SFT). This section not only validates the practical utility of our metric for improving model training but also provides causal evidence linking training data, internal oscillations, and behavioral outcomes.
> >
> > 1. Informing Practical Improvements in Training (Addressing "Lack of Validation" and "Practical Utility") We demonstrate that our framework provides a principled guideline for data selection, a critical aspect of multimodal training and alignment.
> >    - Failure of Easy-to-Hard Generalization: Our experiments reveal a critical phenomenon where models trained solely on "easy" samples (where the target modality is already dominant) fail to generalize to complex scenarios.
> >    - Guidance for Data Selection: We show that data efficiency is strictly governed by the reasoning difficulty identified by our framework. To achieve robust preference alignment, it is essential to move away from trivial samples and prioritize boundary cases (ambiguous and hard-negative regions) identified by our relative uncertainty metric. This directly answers how our framework informs training: it acts as a filter to select the most effective data for alignment tasks like SFT or DPO.
> >
> >
> > 2. Causal Link Between Training, Oscillation, and Behavior (Addressing "Interpretability Gap") We go beyond correlation by showing how specific training interventions causally impact internal oscillations and, consequently, model decisions.
> >
> >    - Mechanism of Steering: As shown in Figure 7b, we tracked the change in oscillation frequency after fine-tuning. We found that successful steering (training on hard/ambiguous data) leads to a substantial reduction in oscillations (negative bars), signaling that the model has confidently committed to the new preference.
> >
> >    - Causal Evidence: Conversely, training on "easy" data fails to suppress this instability. For instance, models trained only on easy samples exhibited the poorest oscillation control in hard regions (e.g., a +164% spike in oscillation for LLaVA), directly resulting in poor downstream performance. This establishes a clear causal chain: the difficulty distribution of training data determines the stability of the internal decision process (oscillation), which in turn dictates the robustness of the final modality-following behavior.
> >
> > Section 6 provides concrete evidence that our framework is not just a theoretical tool but a practical guide for optimizing data selection in model alignment, effectively bridging the gap between internal interpretability and downstream performance.

---

> ### Author Response · Authors · 2025-11-23
> **Reply to Reviewer VpJP on Q2**
>
> **[Q2]**
>
> > The experiments mainly validate the correlation between H^{(t)} (text entropy) and accuracy under unimodal (T, Q) inputs. However, this only demonstrates that entropy reflects uncertainty in isolated textual reasoning. How does this directly justify its use as the governing variable for modality-following behavior in the conflict setting, where cross-modal interactions and contextual dependencies might differ?
>
> We justify using unimodal entropy as the governing variable based on three aspects:
>
> 1. Theoretical Justification: Decomposing "Signal Strength" from "Conflict Outcome"
>    - Our framework aims to explain why a model chooses one modality over another. To do this causally, we must treat the conflict as a competition between two distinct information streams. Independent Variables: Unimodal entropy ($H^{(t)}$ and $H^{(v)}$) serves as a measure of the model's independent confidence in each modality's "raw signal" before the interference occurs. This reflects the intrinsic reasoning capability for that specific input. Avoiding Confounding Factors: If we were to use multimodal entropy (uncertainty during the conflict), we would be measuring the consequence of the conflict (the resulting confusion) rather than its cause. By using unimodal metrics, we effectively disentangle the input quality (reasoning difficulty) from the interaction dynamics, allowing us to predict how the model resolves the clash based on the comparative strength of the unimodal priors.
> 2. Behavioral Regularity
>    - As shown in Figure 4, when we order cases by their relative unimodal uncertainty ($\Delta H_{rel}$), a smooth, monotonic modality-following curve emerges across all models and datasets 1.Conclusion: The existence of this strict pattern proves that, despite the complexity of cross-modal interactions, the model's final decision is indeed overwhelmingly governed by the confidence gap between its unimodal reasonings. If contextual dependencies significantly overrode this unimodal foundation, we would expect a chaotic or uncorrelated distribution, not the precise monotonic law we observe.
> 3. Robustness: Capturing the Abstract Concept of Uncertainty (Appendix F.4)
>    - To justify that our framework captures the underlying *cognitive state* governing modality following, rather than being an artifact of the specific entropy formula, we validated the generalization of our metrics in **Appendix F.4**.
>         - **Metric Agnosticism:** We replicated our analysis using alternative uncertainty metrics, including **Prediction Margin** (capturing conflict intensity) and **Negative Log-Probability** (reflecting raw surprisal).
>           - **Universal Law:** As illustrated in **Figure 16**, the monotonic law holds robustly across all these metrics. This confirms that our findings are not tied to the mathematical formulation of entropy but are governed by the model's fundamental perceived uncertainty.
>
> So unimodal entropy is a justified governing variable because it defines the baseline reasoning difficulty (the lower bound) that persists and drives the decision-making process even amidst cross-modal interference.

---

> ### Author Response · Authors · 2025-11-24
> **Updated PDF & Rebuttal Submitted**
>
> Our rebuttal and updated PDF are now available. Please let us know whether the response resolves your concerns, and if so, whether it affects your score—or if there are remaining questions to address.

---

### Author Response · Authors · 2025-11-23
**Summary of Major Revisions: Scenario Expansion, Parallel Experiments, and Downstream Application**

## Summary of Major Revisions: Scenario Expansion, Parallel Experiments, and Downstream Application

We have uploaded a revised manuscript incorporating significant new experiments to address concerns regarding scenario diversity, model generality, and practical utility. **We highlighted all the changes in blue in the PDF.** Below is a summary of the key updates:

**1. Expansion of Real-World Scenarios and Model Diversity (Main Text: Section 4)**
To improve ecological validity, we have moved beyond synthetic datasets:

- **New Real-World Datasets:** We integrated the **$MC^2$ Benchmark** (Color & Object Recognition) and a conflict-modified **Pascal VOC Dataset** (Object Recognition).
- **New Model Families:** We expanded our evaluation to **8 models** by adding **CogVLM2** and **GLM-4V**.
- **Result:** We verify that the "Relative Uncertainty Law" holds robustly across these complex semantic tasks and diverse architectures.

**2. Extensive Parallel Experiments for Robustness (Appendix F)**
We have added **Appendix F** containing comprehensive parallel analyses to demonstrate the universality of our findings:

- **Behavioral Consistency (App. F.1 & F.2):** Detailed breakdowns of performance and uncertainty distributions for *all* 8 models on the new real-world datasets.
- **Robustness Checks (App. F.3 & F.4):** Validation of the monotonic law under varying **Absolute Difficulty** (High vs. Low Entropy) and using **Alternative Metrics** (Prediction Margin, Negative Log-Probability).
- **Universality of Internal Mechanism (App. F.5):** Layer-wise probing extended to **all models and datasets**, confirming that the "oscillation" mechanism in ambiguous regions is a universal property of MLLMs.

**3. New Application Section: Preference Steering (Main Text: Section 6)**
To demonstrate practical utility, we added **Section 6**, applying our framework to **Supervised Fine-Tuning (SFT)**. We show that:

- Training on "easy" samples fails to generalize, whereas targeting **boundary cases** identified by our uncertainty metric is essential for robust alignment.
- Effective steering causally reduces internal oscillations, making models more decisive.

We believe these revisions provide strong evidence for the generalizability and practical value of our framework.

---

### Author Response · Authors · 2025-12-03
**TL;DR A Summary of Discussion by Authors**

Dear Reviewers, AC and Researchers,

Thank you all for your effort at this difficult time to our ICLR community. We are grateful that the reviewers help us to improve our work. We are encouraged that 1 of 4 reviewers raised their scores before 26 Nov, from 6644 to 8644. We have supplemented amounts of experiments, analyses and clarifications to strengthen our work. In the revised pdf, we added the description to all extension experiments during rebuttal period in **Section 4, Section 5, Section 6, and Appendix F**, which addressed most of the reviewer's concerns.

We first recap the major concerns from reviewers:

**[Generalizability and Benchmarks]:** Concerns about the reliance on synthetic datasets ("toy-level") and limited model diversity. (W1@VpJP, W2@x1XN, W1&W3@yL48, W1@VBB8)

**[Metric Robustness]:** Concerns about relying solely on Entropy as a proxy for uncertainty and requests for alternative metrics. (Q2@VpJP, W2@yL48, W3@VBB8)

**[Practical Utility]:** Questions regarding the application of the framework to downstream tasks like preference steering or alignment. (W2@VpJP, W4@yL48)

**[Mechanism and Analysis]:** Clarifications on the internal oscillation mechanism, layer-wise distribution, and calculation details. (W3@VpJP, Q1@x1XN, Q2@yL48)

Now we summarize our discussion, and we trace the score changes below:

- **Reviewer VBB8, raised the score from 6 to 8 , at 2025-11-23 07:57 AoE.**

  - **W1.** We addressed the "ecological validity" concern by extending our evaluation to real-world photography benchmarks ($MC^2$ and Pascal VOC), showing the domain-agnostic nature of our findings (**Section 4**).

  - **W3.** We conducted robustness checks using alternative uncertainty quantification methods (Prediction Margin, NLL) in **Appendix F.4**, proving our conclusions are not artifacts of the entropy formula.

  - **W2.** We added a discussion on prior interpretability frameworks (e.g., "Vision-for-language"), bridging the gap between static capability analysis and our dynamic decision-making framework.

- **Reviewer x1XN, original score 6, no response received**

  - **W2, W3 & Q6.** We conducted extensive parallel experiments for all 8 models (including LLaVA, Qwen, CogVLM2, GLM-4V) in **Appendix F**, confirming that the behavioral patterns and internal mechanisms are universal and not specific to selected models.

  - **W1.** We expanded the **Related Work** to discuss linguistic priors and semantic bias, clarifying the connection between static priors and our dynamic reasoning framework.

  - **Q1.** We clarified the calculation of "Average Concept Oscillations" and validated the metric's reliability across new datasets in **Section 5** and **Appendix F.5**.

- **Reviewer VpJP, original score 4, no response received**.

  - **W1.** We significantly expanded our experimental scope to include challenging real-world datasets ($MC^2$ and Conflict-Pascal VOC) and added new model families (CogVLM2, GLM-4V), proving that our findings hold in complex scenarios.

  - **W2 & Q1.** We added a new **Section 6 (Application)**, demonstrating that our framework can guide data selection for Supervised Fine-Tuning (SFT). Experiments showed that training on "ambiguous" data identified by our metric causally reduces internal oscillations and improves steering.

  - **Q2.** We validated the robustness of our metrics in **Appendix F.4**, showing that the "Relative Uncertainty Law" holds using alternative metrics like Prediction Margin and Negative Log-Probability.

- **Reviewer yL48, original score 4**

  - **The reviewer acknowledged our clarifications and explicitly stated they had "no further questions," but maintained the original evaluation without providing additional technical justification.**

  - **W1 & W3.** We supplemented the $MC^2$ and Pascal VOC benchmarks and included CogVLM2 and GLM-4V (Section 4), addressing the concern about "toy datasets" and model diversity.

  - **W2.** We addressed the entropy robustness concern by replicating our analysis with Prediction Margin and NLL (Appendix F.4), confirming metric independence.

  - **W4.** We demonstrated the practical utility of our framework in **Section 6**, showing how relative uncertainty guides effective data selection for preference steering.

  - **Q2.** We provided a detailed layer-wise analysis (**Figure 5b, Appendix F.5**), confirming that oscillations are strictly localized to middle-to-late layers, reflecting high-level cognitive conflict resolution.

Authors

---

### Meta-Review · Area_Chair_CfNN · 2026-01-06

**Summary:**

The paper introduces a framework to analyze "modality following" in Multimodal LLMs by decomposing the behavior into relative reasoning uncertainty and inherent modality preference. During the discussion phase, the authors conducted a comprehensive rebuttal, successfully adding real-world benchmarks (Pascal VOC, MME) and alternative uncertainty metrics, which convinced one reviewer to raise their score to an 8. However, the majority of reviewers maintained their reservations, preventing a consensus for acceptance. The primary grounds for rejection are the limited novelty of the core insight—that models follow the modality they are more confident in—which was viewed as fundamentally intuitive. Furthermore, despite the additional experiments, skepticism persists regarding the ecological validity of the "universal laws" derived largely from synthetic data, as well as the limited practical utility of the framework for improving downstream model performance compared to existing alignment techniques. Thus, this paper was not recommended for acceptance at its current form.

**Reviewer Concerns:**

Addressed Concerns: (1) Dataset Scope: The authors successfully expanded the evaluation from synthetic shapes to real-world benchmarks (MME, Pascal VOC) and included more model families (CogVLM2, GLM-4V), addressing the specific requests of Reviewers VpJP, yL48, and VBB8. (2) Metric Robustness: The authors demonstrated that their findings hold when using metrics other than entropy (e.g., Prediction Margin, NLL), addressing the technical concerns of Reviewers yL48 and VBB8. (3) Internal Mechanisms: The authors provided parallel analyses of the oscillation mechanism across different models, addressing Reviewer x1XN’s request for robustness checks.

Outstanding Concerns: (1) Practical Impact: Reviewer VpJP (and implicitly yL48) questioned the downstream utility. While the authors added a steering experiment, it was not sufficient to shift the consensus that the work is primarily an interpretability study with limited immediate application for improving state-of-the-art models. (2) Generalizability: Despite the new data, the "cleanliness" of the laws observed in the synthetic setting may not fully translate to the nuance of broad-domain multimodal tasks, leaving the reviewers (particularly yL48) skeptical of the "universal law" claim. (3) Intuitive Nature: The core finding—that uncertainty drives preference—is fundamentally intuitive, which limits the perceived technical depth despite the rigorous execution.

**Reviewer Scores:**

Reviewer VpJP: 4 $\rightarrow$ 4 (Unchanged)

Reviewer x1XN: 6 $\rightarrow$ 6 (Unchanged)

Reviewer yL48: 4 $\rightarrow$ 4 (Unchanged)

Reviewer VBB8: 6 $\rightarrow$ 8 (Increased)

---

### Decision · Program_Chairs · 2026-01-26

Reject